# Mapping age and basal conditions of ice in the Dome Fuji region, Antarctica, by combining radar internal layer stratigraphy and flow modeling

Zhuo Wang[1,2], Ailsa Chung[3], Daniel Steinhage[1], Frédéric Parrenin[3], Johannes Freitag[3], and Olaf Eisen[1,4]

[1]Alfred-Wegener-Institut, Helmholtz-Zentrum für Polar und Meeresforschung, Bremerhaven, Germany
[2]College of Geo-Exploration and Technology, Jilin University, Changchun, China
[3]Univ. Grenoble Alpes, CNRS, IRD, IGE, Grenoble, France
[4]Department of Geosciences, Universität Bremen, Bremen, Germany

**Correspondence:** Olaf Eisen (olaf.eisen@awi.de)

**Abstract.** The Dome Fuji (DF) region in Antarctica is a potential site for an ice core with a record of over one million years. Here, we combine large-scale internal airborne radar stratigraphy with a 1-D model to estimate the age of basal ice in the DF region. The radar data used in the study were collected in a survey during the 2016–2017 Antarctica season. We transfer the newest age–depth scales from the DF ice core to isochrones traced in radargrams in the surrounding 500 km × 550 km region. At each point of the survey the 1-D model uses the ages of isochrones to construct the age–depth scale at depths where dated isochrones do not exist, the surface accumulation rate and the basal thermal condition, including melt rate and the thickness of stagnant ice. Our resulting age distribution and age density suggest that several promising sites with ice older than 1.5 million years in the DF region might exist. The deduced melt rates and presence of stagnant ice provide more constraints for locating sites with a cold base. The accumulation rates range from 0.015 to 0.038 m a$^{-1}$ ice equivalent. Based on sensitivity studies we find that the numbers of picked isochrones and the timescale of the ice core severely affect the model results. Our study demonstrates that constraints from deep radar isochrones and a trustworthy timescale could improve the model estimation to find old ice in the DF region.

## 1 Introduction

In order to understand the Quaternary climate, the progression of glaciations and the carbon cycle, we need to find continuous and undisturbed ice-core records back to 1.5 million years BP (before present, present defined as 1950) (Fischer et al., 2013; Jouzel and Masson-Delmotte, 2010). The switch from more regular 41 ka glacial cycles (1500 to 1200 ka BP) to current 100 ka glacial cycles, which occurred in the time interval between 1200 ka and 900 ka BP, is known as the mid-Pleistocene transition (MPT) and is still not fully understood (Lisiecki and Raymo, 2005). $CO_2$ and other greenhouse gas may have either forced this switch or might have responded to it (Willeit et al., 2019). A direct record of greenhouse gases with atmospheric record covering this period can only be found in Antarctica ice cores (Fischer et al., 2013). Moreover, isotopic and chemical records in ice cores of that age can provide additional information on temperature, ice dynamic changes and magnetic reversals, which

can be analyzed together with other marine and terrestrial records (Raymo et al., 2006; Raisbeck et al., 2006; Singer and Brown, 2002). Hence, identifying "Oldest Ice" sites in Antarctica is one of the primary challenges for ice-core research.

It is a huge challenge to retrieve continuous old ice-core records, as the oldest ice is compressed in deep layers near the base of the ice sheet. Ice older than one million years could have melted away by reaching the pressure melting point or be disturbed, and thus not useful for ice-core analyses, because of complicated processes in the deepest ice (Van Liefferinge and Pattyn, 2013). As one consortium in the International Partnerships in Ice Core Sciences (IPICS), the European "Beyond EPICA–Oldest Ice" (BE–OI) consortium initiated pre-site surveys in the wider Dome Concordia (DC) and Dome Fuji (DF) regions. Van Liefferinge and Pattyn (2013) evaluated potential sites of million year-old ice in Antarctica considering ice velocity, ice thickness and geothermal heat flow (GHF) based on a thermodynamical model. In a follow-up study, Van Liefferinge et al. (2018) focused on more detailed sites for oldest ice in the DF and DC regions, using more robust criteria (for ice thickness and velocity), and a metric for the shape of the bed. In the DC region, Young et al. (2017) extended ice thickness coverage, mapped the basal roughness and found more subglacial lakes through high-resolution aerogeophysical surveys, and finally assessed the previous old ice candidates. Parrenin et al. (2017) inferred the age of ice based on 1-D thermo-mechanical model constrained by radar observations and identified two target areas where ice older than 1.5 Ma may exist. Lilien et al. (2021) refined the age–depth scale at Little Dome C (LDC), 40 km from the DC site, by adapting a 1-D ice flow model constrained by high-resolution radar data collected around the drill site selected for the Beyond EPICA project. They suggested 1.5 Myr old ice exists at $\sim 2500$ m depth, where stratigraphy is still intact and preserved with analysable resolution. In the Dome A region, Sun et al. (2014) estimated ice age around Kunlun station by applying a three-dimensional, thermomechanically coupled full-Stokes model, which indicated that in the area without basal melting the ice age at 95 % depth could be limited to 1.5 Ma. Beem et al. (2021) suggested that Titan Dome is an unlikely to have old ice covering MPT based on depth distribution of dated internal horizons traced in the radar data, age modeling constrained by radar horizons and faster flow that ceased during the last glacial maximum.

The Dome Fuji region is a potential area for holding oldest ice in Antarctica. The DF drill site (77°19′01″ S, 39°42′12″ E) (Ageta et al., 1998) is located at an elevation of 3810 m, with an ice thickness of 3028±15 m (Fujita et al., 1999, 2015), an annual mean air temperature of −54.4°C (Kameda et al., 2009) and a mean annual accumulation of $\sim 24$ mm w.e.a$^{-1}$ (Fujita et al., 2011). The first deep ice core at DF, which was drilled from 1995 to 1996, reached 2503.52 m and covered the past $\sim 340$ ka using the isotopic $\delta^{18}O$ record for dating (Ageta et al., 1998; Watanabe et al., 1999). Kawamura et al. (2007) used $O_2/N_2$ measurements, a proxy of local summer insolation, to build a new timescale (referred to as DFO-2006). Based on these $O_2/N_2$ age markers, Parrenin et al. (2007) used a 1-D ice flow model to reconstruct the timescale down to the ice near the base and accumulation rates (referred to as DFGT-2006). During the austral summers from 2003/04 to 2006/07, the second deep ice core, only 48 m away from the first one (Saruya et al., 2022), was finally drilled to a depth of 3035.22 m. It is considered to be very closed to the bedrock (Motoyama, 2007; Motoyama et al., 2021) and the temperature at the bottom of this borehole reached the melting point (Talalay et al., 2020). By synchronising the isotopic $\delta^{18}O$ record of the DF ice core with that of the ice core at DC (AICC2012), Dome Fuji Ice Core Project Members (2017) dated the DF deep ice core back to $\sim 720$ ka and

deduced accumulation rates. A timescale which combines DFO-2006 (< 342 ka) and AICC2012 (> 344 ka) was then proposed (referred to as DFO2006+AICC2012) (Dome Fuji Ice Core Project Members, 2017).

In addition to the direct analysis of ice-core proxies, some ice models were applied in the larger DF region to investigate basal thermal states and age fields. Seddik et al. (2011) adapted a three-dimensional, thermomechanically coupled ice flow model with induced anisotropy to a $\sim$ 200 km$\times$200 km domain around the DF drill site. They simulated a basal melt rate of $\sim$ 0.35 mm a$^{-1}$ at DF, and found that the consideration of anisotropy would decrease the inferred age of the ice. Karlsson et al. (2018) presented an updated subglacial topography with a resolution of 10 km in the DF region based on airborne radar surveys conducted by the Alfred Wegener Institute, Helmholtz Centre for Polar and Marine Research (AWI), as part of the Beyond EPICA project. With new bed topography, they refined some promising oldest ice sites proposed by Van Liefferinge and Pattyn (2013) using the same model and suggested that especially the region immediately south of Dome Fuji station is promising for holding old ice. A 1-D ice flow model called IcIES-2 was adapted to DF and the DF-New Dome Fuji site (NDF) transect by Obase et al. (2023). They examined the influence of ice thickness and GHF on the age of the ice and pointed out that ice thickness is one of the most critical factors for the preservation of old ice. They suggested that analyzable 1.5 Ma old ice could be expected at DF when the GHF is small enough to keep the basal ice from melting.

Tsutaki et al. (2022) compiled a new ice thickness data set collected by ground-based radar surveys over the last 30 years, which revealed higher resolution of the complex landscapes compared with the previous data sets. Based on the new compilation, they examined roughness and slope of the ice–bed interface, the stress state of the ice and the subglacial hydrological conditions in the vicinity of DF and NDF, thus provided a substantial set of constraints for identifying old ice candidate sites.

We present another method to complement the progress already made by constraining age and the basal thermal condition in the larger DF region (i.e. roughly a 500 km $\times$ 550 km perimeter) by isochrones detected by radar. We connect the ice-internal isochrone stratigraphy in the larger DF region to the DF drill site through isochrones traced in the airborne radar data collected during the 2016–17 survey conducted by AWI. We apply a 1-D ice flow model (see more details of the model in the companion paper of Chung et al. (2023)) to determine the age–depth scale below the available isochrone stratigraphy, accumulation rates and also to derive either melt rates or the thickness of the stagnant ice in the DF region. We finally discuss the reliability of the results, conduct sensitivity experiments to quantify the effect of the number of used isochrone and the timescale of the ice core on our age estimates as well as the other modeling results.

## 2  Data and Methods

### 2.1  Data collection

We acquired 26 radar profiles with the AWI airborne radio-echo sounding (RES) system operated on the Basler BT-67 aircraft Polar6 (Wesche et al., 2016) during the 2016/17 Antarctic season. The radar survey was conducted from a temporary camp (79° S, 30° E) 290 km south-west from the DF station. Survey lines have parallel line spacing of 10 km in the northern part of the study area and 15 km line spacing in the southern part with smaller spacing distances when approaching and leaving the camp (Karlsson et al., 2018). Of the 26 profiles available we analyse 22 with lengths varying from 622 km to 898 km. The

study area covers a region of about 270,000 km$^2$ and an elevation range of about 3400 m–3810 m (Fig. 1). In this region, ice
surface velocities range from 0 to 5.8 m a$^{-1}$.

The AWI RES system transmits radar waves with a center frequency of 150 MHz, a band width of 20 MHz and an amplitude
of 1.6 kW. During the survey it effectively operated as a pulse-limited radar with the 600 ns wide pulse. The theoretical vertical
resolution in ice for the 600 ns burst is 50 m (Nixdorf et al., 1999). In this study we use radar returns of the 600 ns long burst
from internal reflection horizons (IRHs) as well as from the ice–bedrock interface. The raw radar data has a mean spacing
of 5 m along the flight line (which varies with real speed and direction of aircraft) and is sampled at a time interval of 4 ns
(Karlsson et al., 2018).

Before picking the IRHs and the ice–bed interface, the radar data are resampled to 12 ns and stacked 7 fold, which leads
to a mean trace spacing of $\sim$ 35 m. In addition, a low-pass filter and a running average filter are used to decrease noise.
Automatic gain control (AGC) is used to balance the gain and facilitate horizon tracing. Processing is performed in the seismic
environment of the Echos software from Paradigm Geophysical.

## 2.2 Horizon picking and dating

In order to provide age markers (i.e., the age of IRHs) and ice thickness to use as constraints in our 1-D flow model, IRHs
are traced in the two-way travel time (TWT) domain. The surface reflection is picked automatically in the program "Echos"
and then subtracted in all traces to shift the first break of the radar data to time zero. The maximum reflection power of IRHs
is traced manually in the seismic software package "Section", which allows IRHs to be continuously traced in different radar
profiles through intersections between profiles. This ensures that the same isochronous horizons are traced. We trace 6 (H1,
H2, H4–H7) or 7 (H1–H7) relatively distinct and continuous IRHs in the radar profiles, since the third IRH H3 is not clear and
continuous enough to be traced in some profiles. Ice–bed returns were picked by Karlsson et al. (2018) through semi-automatic
detection routines in MATLAB. These ice thickness data are available on PANGAEA (Eisen et al., 2020). The ice–bed returns
are often diffuse, especially around mountain peaks, which results in disagreements when using different methods to trace ice–
bed returns, and thus differences in ice thickness and modeling results. We emphasize that Karlsson et al. (2018) picked the
first (uppermost) ice–bed return when there were uncertainties, so that ice thickness estimates can be considered a minimum
ice thickness in some places.

TWT is converted to depth assuming a constant radar wave speed of 168.5 m μs$^{-1}$ in ice (Winter et al., 2017; Lilien et al.,
2021) and a 15.5 m firn correction calculated from depth-density curve in the B53 ice core. The B53 ice core was drilled to
202 m depth by the AWI team during the survey period and is located at 79°47′38″ S, 31°54′19″ E, and $\sim$ 203.5 km away from
the DF drill site (Fig. 1). The point of closest approach to the DF drill site on our radar profiles is located $\sim$ 91.1 m away, at
77°19′ S, 39°41′59″ E, on the profile 20170240. Ice thickness at this nearest point observed in the radargram is about 3044.8 m,
and the ice thickness at the DF drill site interpolated between our radar observations is about 3050.5 m. This corresponds within
the the uncertainty estimates with a previously inferred ice thickness of 3028±15 m from a radar observation (Fujita et al.,
1999), and it approximates the depth of the second DF deep ice core, 3035.2 m, which is considered to be very close to ice–bed
interface (Motoyama, 2007).

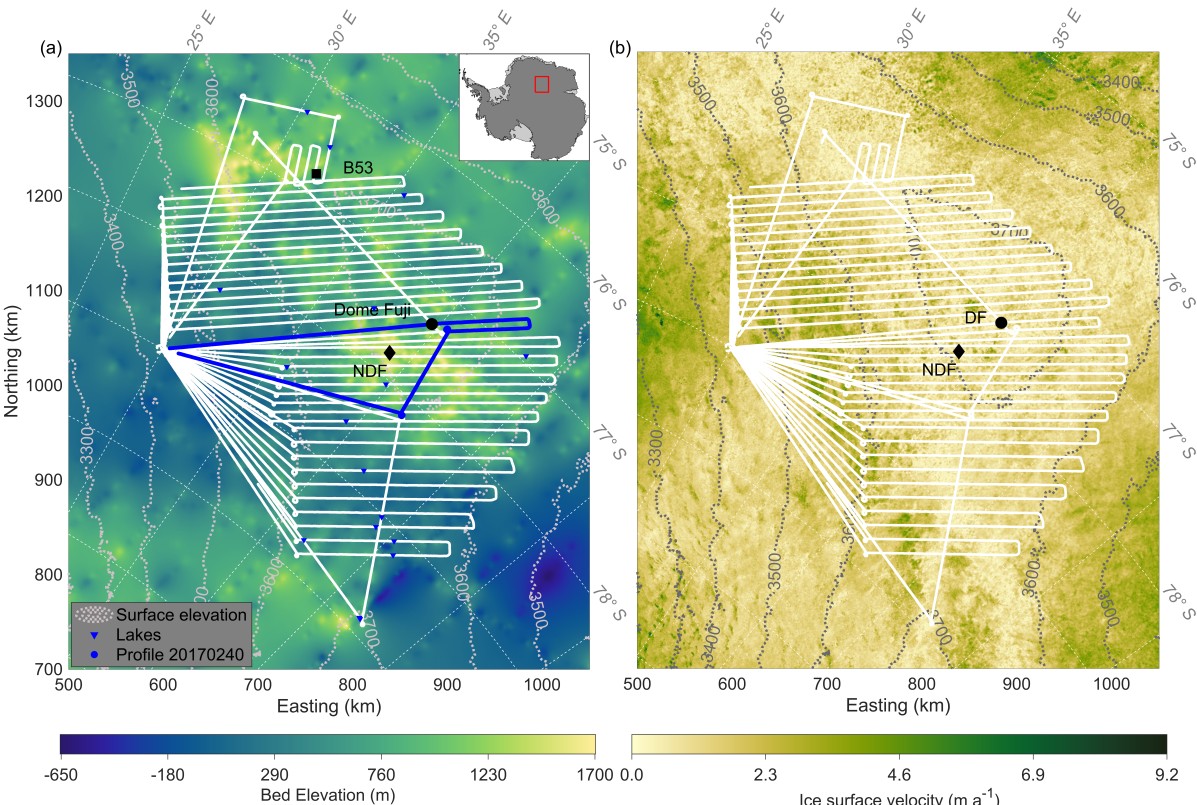

**Figure 1.** (a) Study area in the DF region, with inset showing the position in Antarctica. The white lines represent the radar survey profiles used in our study. The blue line shows the example profile 20170240. We use surface elevation (contour map) and bed elevation from Morlighem et al. (2020) and Morlighem (2022). Subglacial lakes were identified by Karlsson et al. (2018). (b) Ice surface velocities mapped by (Rignot et al., 2017, 2011; Mouginot et al., 2012, 2017).

IRHs at this location closest to the DF drill site are firstly converted to depth and then dated by vertical interpolation of the ages from the DFO2006+AICC2012 timescale (Dome Fuji Ice Core Project Members, 2017), available to a depth of 3031 m with an ice age of 715.9 ka BP, and finally transferred to the radargram at the depths of the IRHs. Fig. 2 shows traced IRHs H1–H7 in the profile 20170240 with the point of closest approach shown as white vertical line. The age of IRHs, ranging from 31.4 ka (H1) to 169.1 ka (H7), and their age uncertainty, are marked in Fig. 2. Ages of different IRHs are then transferred to all profiles via our network, which then serve as the primary constraint to the 1-D model.

## 2.3 Age uncertainty of internal reflection horizons

The uncertainty of IRH age estimates directly impacts the reliability of the results from the model, as it includes uncertainty of the ice-core agescale and age uncertainty caused by depth uncertainty of IRHs. For the depth uncertainty of IRHs, we consider slope-induced uncertainties caused by the offset of the closest radargram to the DF drill site, of the firn correction, of the dielectric constant of ice and of the range precision of the estimate in determining depth (Cavitte et al., 2016).

The slope of each IRH varies from 1 m km$^{-1}$ to 14.7 m km$^{-1}$, which results in a corresponding uncertainty from 0.1 m to 1.5 m for the 91.1 m offset (the distance between the DF drill site and the point of closest approach on the radar profile). For the firn correction, we used AWI's ICE-CT system to measure the density–depth profile in the upper 126 m of the B53 ice core, with an observational error up to 1 % (Freitag et al., 2013). This results in an uncertainty of 0.5 m in the firn correction. The depth uncertainty of dielectric constant affected by anisotropy and temperature is taken to be 1 % (Lilien et al., 2021).

The estimate of the range precision is always numerically smaller than the vertical resolution, which is determined by the pulse width of the radar waveform, the signal-to-noise ratio (SNR) and the sub-resolution reflector fluctuations. The last term could be ignored when the reflectors display a continuity in reflection amplitudes and consequently traceability (Cavitte et al., 2016). In our case, the precision of the range estimate is calculated from the SNR and range resolution. In our case the precision of the range estimate is almost the same as the range resolution ($\sim$ 25 m), but the precision of the range estimate is actually lower ($>$50 m) in the deep ice since our radar system has a 50 m pulse width.

The resolution of our system is lower than that of more advanced radar systems, and this causes a smaller number of internal horizons as well as a lower traceability. Moreover, the bedrock topography is characterized by a series of mountain ranges and valleys and wide melting distribution in the Dome Fuji region, which leads to the discontinuity of isochrones at some places, especially near the bottom. Therefore, we need to consider the sub-resolution of different reflectors in our analysis.

We find that the uncertainty caused by the low traceability and continuity is large when we trace the horizons manually. We therefore attempt to trace horizons via several path to circumvent locations where horizons are disturbed or discontinuous. Therefore our best guess for the uncertainty of each IRH depth is based on continuity and definition during manual tracing. It varies from 20 m to 50 m.

The overall depth uncertainty varies from 28.5 m to 70.5 m, leading to an age uncertainty range from 2.1 ka to 16.8 ka. The age uncertainty of the ice core itself is interpolated from the age errors of the agescale DFO-2006 (Kawamura et al., 2007). In total, the age uncertainties range from 3.0 ka to 19.0 ka for the 7 IRHs.

## 2.4 1-D age model

To extrapolate the age–depth profile in the study area below the depth of the deepest IRH, we use a 1-D pseudo-steady ice flow model developed by Parrenin et al. (2006, 2017) but with simplified constraints. This model assumes that the geometry, the shape of the vertical velocity profile and the relative density profile are constant. The real ice age $\chi$ can be calculated using the

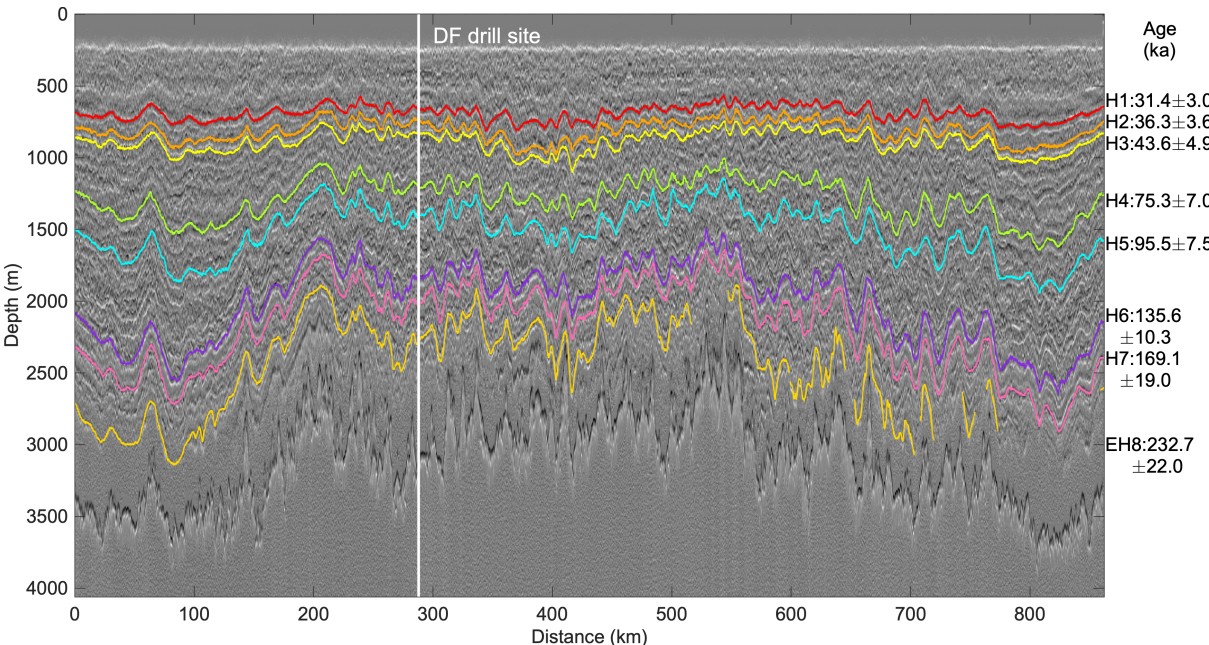

**Figure 2.** Radargram of the profile 20170240. The vertical white line shows the location of the DF drill site. Lines with different colours represent continuous horizons H1–H7 and the specially traced discontinuous horizon EH8. The dated age and age uncertainty of each horizon is marked on the right.

steady age $\bar{\chi}$ and the temporal factor $r(t)$ by

$$\bar{\chi} = \int_0^t r(\chi')\mathrm{d}\chi', \tag{1}$$

where $r(t)$ is deduced from the accumulation record of the DF ice core. We assume $r(t) = 1$ beyond the extent of the DF ice-core record (715 ka BP), else

$$r(t) = \dot{a}(x,t)/\bar{a}(x), \tag{2}$$

where $\dot{a}$ is the accumulation rate and $\bar{a}(x)$ is the temporally averaged accumulation rate at a certain point $x$. The steady age $\bar{\chi}$ can be inferred from depth $d$ and the layer thickness $\lambda(d)$,

$$\bar{\chi}(d) = \int_0^d \frac{1}{\lambda(d')}\mathrm{d}d'. \tag{3}$$

Assuming that there is no basal melt, $\lambda(d)$, approximated by the Lliboutry model (Lliboutry, 1979), is

$$\lambda(d) = \bar{a}\left(1 - \frac{p+2}{p+1}\left(\frac{d}{H_m}\right) + \frac{1}{p+1}\left(\frac{d}{H_m}\right)^{p+2}\right), \tag{4}$$

where $p$ is a shape factor controlling vertical deformation (Lilien et al., 2021), $H_m$ is the mechanical ice thickness, which is different from the observed ice thickness $H_{obs}$. The main difference between the model we use here and the one developed by Parrenin et al. (2006) is that no thermal modeling and thermal boundary conditions are considered here. Instead, we use the inferred $H_m$ to judge if melting is present or if stagnant (i.e. dynamically irrelevant) ice prevails. When $H_m$ is greater than the observed ice thickness $H_{obs}$, we have melting conditions at the base. Otherwise, there is stagnant ice. If there is basal ice

melting, the melt rate $m$ can be obtained by

$$m = \lambda(H_{obs}), \tag{5}$$

We use a Scipy least square optimization to deduce the age–depth profile by varying $\bar{a}$, $H'_m$ and $p'$, where $H_m = e^{H'_m}$, $p = e^{p'-1}$ to prevent $p < -1$ and $H_m < 0$. The minimized cost function is

$$S = \sum \frac{(\chi_i^{iso} - \chi^{mod}(d_i^{iso}))^2}{(\sigma_i^{iso})^2} + \frac{(p'_{prior} - p')^2}{(\sigma^{p'})^2}, \tag{6}$$

where $i$ is the ordinal number of the IRH, $\chi^{iso}$ is age of the IRH, $\sigma^{iso}$ is the confidence interval on the age, $\chi^{mod}$ is modeled age, and $d^{iso}$ is the depth of the IRH, $p'_{prior} = 3$ and $\sigma^{p'} = 1$ (Parrenin et al., 2007; Chung et al., 2023). The uncertainty of each inverted parameter could be inferred from the optimization and the covariance matrix. To quantify the reliability of the model at each point, we introduced a reliability index $\sigma_R$, i.e., the standard deviation of residuals,

$$\sigma_R = \sqrt{\frac{R^T R}{n_{iso}}}, \tag{7}$$

where $n_{iso}$ is the number of IRHs, $R$ is the residuals deduced by

$$R = \frac{\bar{\chi}^{iso} - \bar{\chi}^{mod}}{\bar{\sigma}_{\chi^{iso}}}. \tag{8}$$

In this way the model achieves the balance of efficiency and numerical requirements. More details on the model can also be found in the companion paper (Chung et al., 2023).

## 3 Results

Age, age uncertainty of IRHs and temporal variations of accumulation rates at the DF drill from Dome Fuji Ice Core Project Members (2017) are used to constrain the 1-D steady-state ice flow model. The output variables of the model are accumulation rate $\dot{a}$, shape factor $p$, mechanical thickness $H_m$, age–depth distribution and either basal melt rate or the thickness of the stagnant ice layer.

### 3.1 Modeling results for an example profile

We integrate 1-D modeling results every 1 km along the example profile 20170240, displayed as a cross section through the ice sheet in Fig. 3, to get the 2-D modeled age–depth distribution. We find ice older than 1 Ma along the profile from $\sim 150$ km

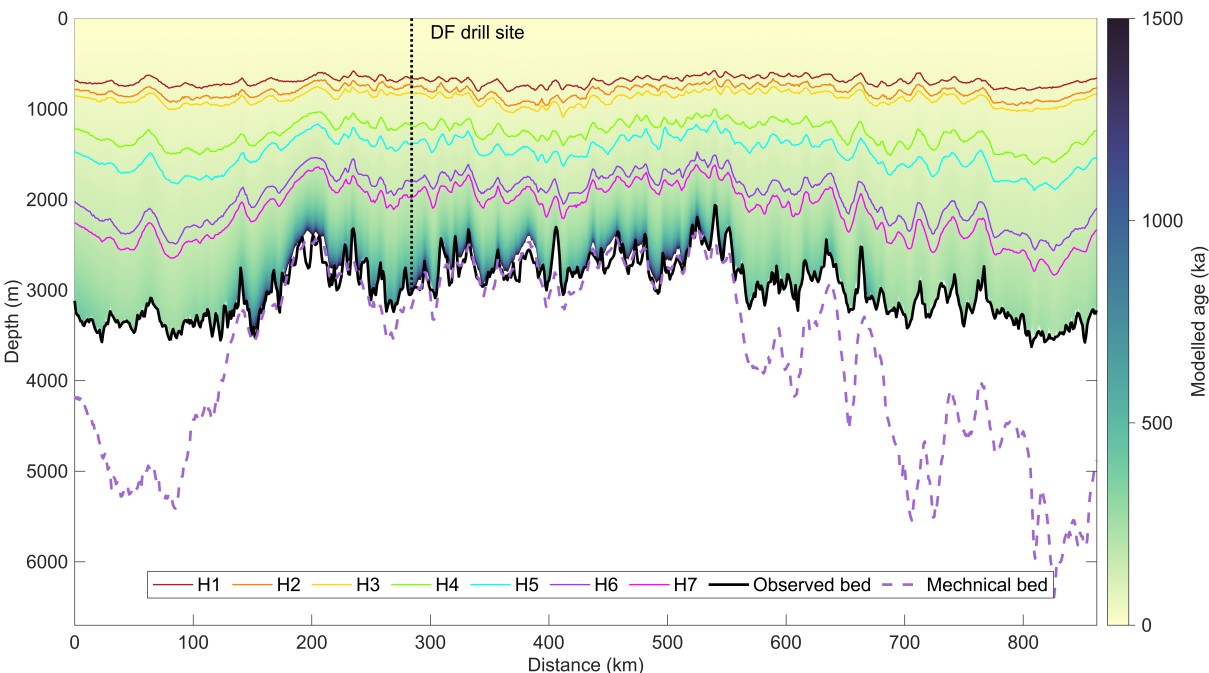

**Figure 3.** Modeled age–depth distribution of the radar profile 20170240. The coloured lines (see legend) correspond to the traced IRHs shown in Fig. 2. The purple dashed line shows the mechanical ice thickness $H_m$ and the black line shows the bed observed in the radargram. Where the purple dashed line is above the black line, stagnant ice is present and the depth difference between the two lines is the thickness of stagnant ice layer. In other cases, melting prevails.

to $\sim$ 550 km, where the ice sheet is thinner than in the other parts. Basal melting is present at the DF drill site and along most parts of the profile, where the mechanical ice thickness $H_m$ (purple dash line) is larger (deeper) than the observed ice thickness (black line).

## 3.2 Age of basal ice

We use 20 ka m$^{-1}$ as a cut-off value for age density of basal ice, beyond which usage of proxies in the ice for paleoclimate reconstruction is currently difficult. This age density corresponds to a full 40 ka climate glacial–interglacial cycle in 2 m of ice. Fig. 4a shows the age of the basal ice (i.e. at the depth of the bed or where the age density reaches 20 ka m$^{-1}$) in the DF region. It varies from 215 ka to 2533 ka. Fig. 4b shows the corresponding depth of the basal ice, which falls in a depth range of 1.6–3.8 km. The age of the basal ice at the DF drill site is extrapolated as 1347.2±503.1 ka at the ice-bed interface. The maximum age of ice at NDF is extrapolated as 1472.1 ka±509.0 ka at depth of 2080.7 m. Fig. 4c shows the age uncertainty of the basal ice, which ranges from 29 to 1050 ka and varies with age of basal ice.

The age density of ice at 1.5 Ma is shown in Fig 5a, with a range of 6–20 ka m$^{-1}$. It is considered sufficient for paleoclimatic reconstructions (Fischer et al., 2013). This figure also points out the four candidate areas where ice of more than 1.5 Ma old

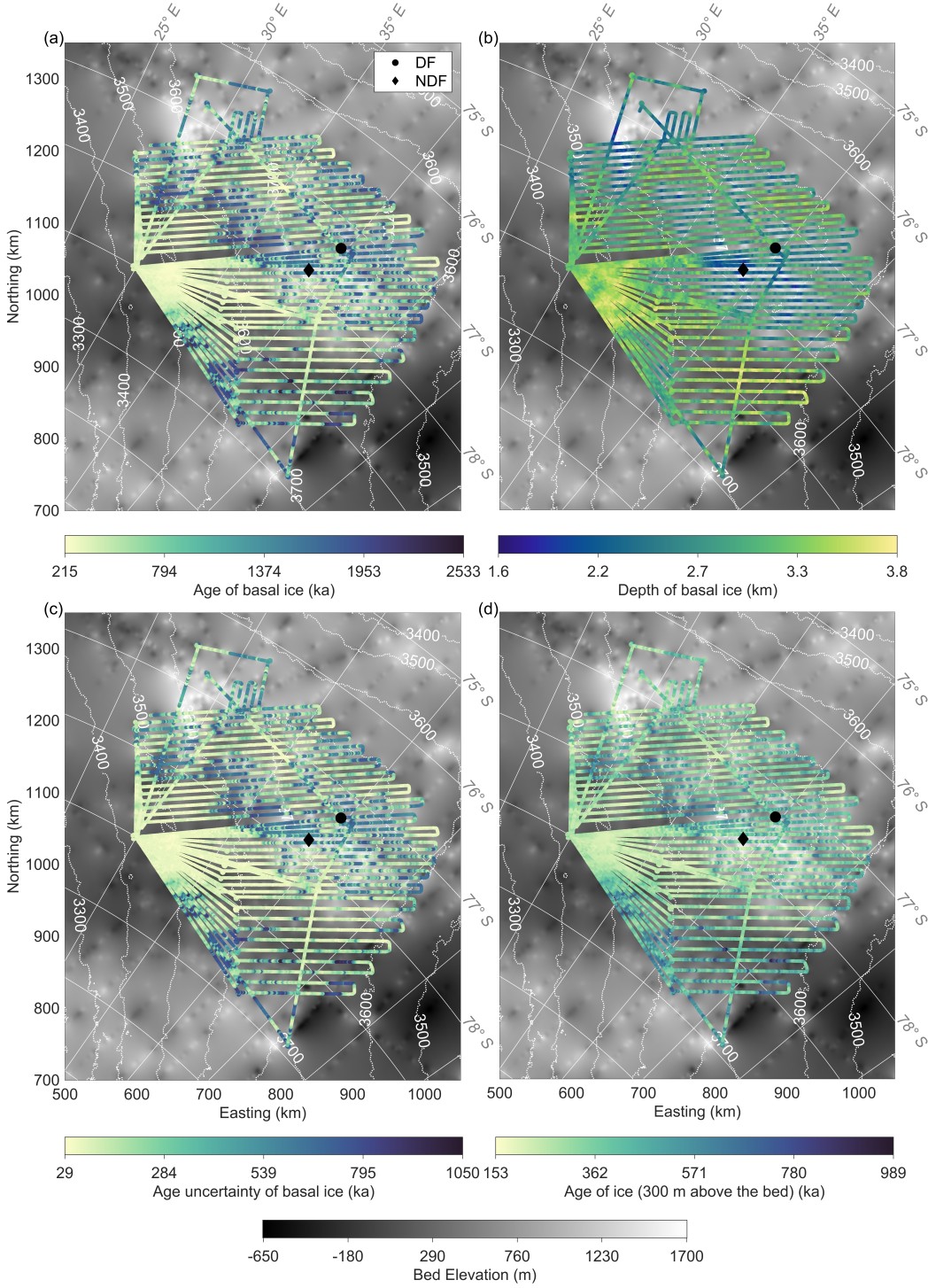

**Figure 4.** (a) Modeled age of the basal ice at a maximum age density of 20 ka m$^{-1}$. (b) Depth of the basal ice at an age density of 20 ka m$^{-1}$. (c) Modeled age uncertainty of the basal ice. (d) Modeled age of the ice at a height of 250 m above the bed.

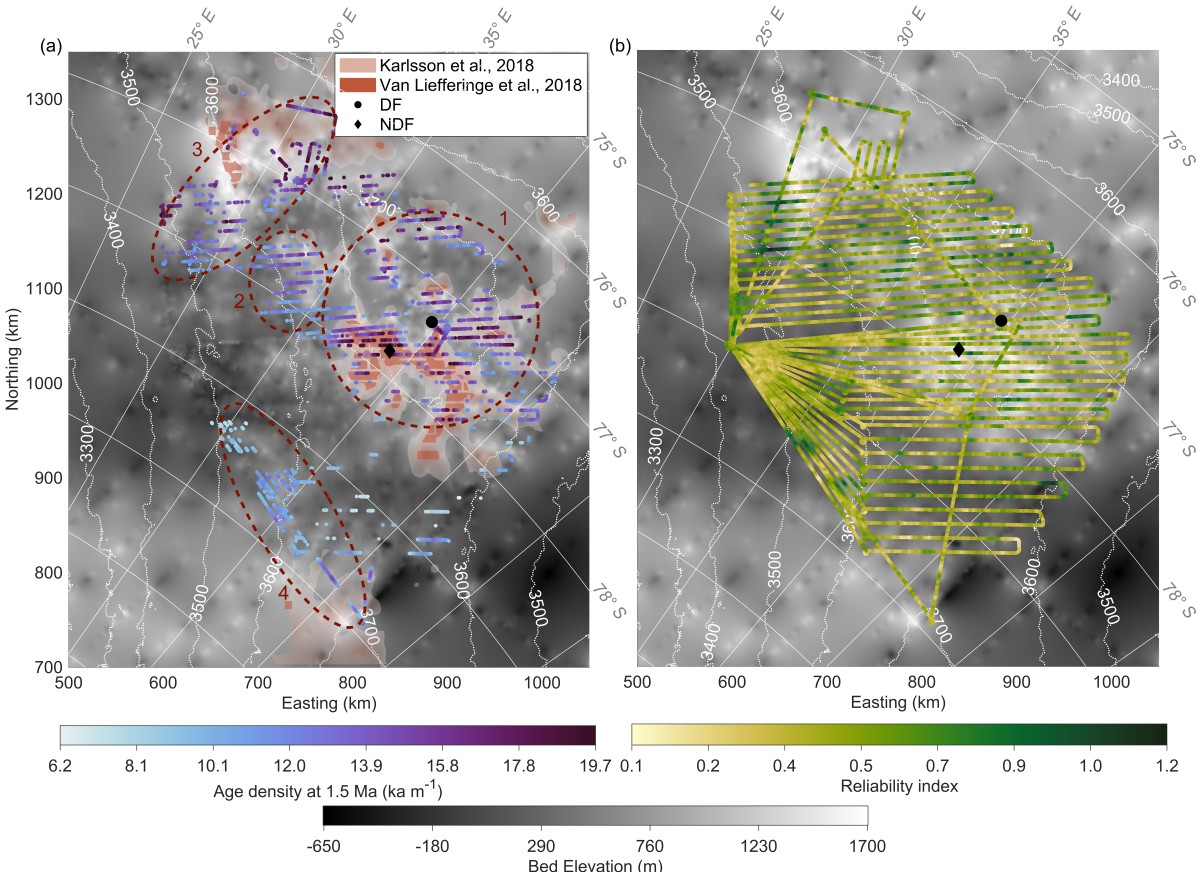

**Figure 5.** (a) Age density of ice at 1.5 Ma, the dark red ellipses show the potential old sites considering basal age and age density. Semi-transparent pink-red colored shades show potential old-ice sites suggested by Karlsson et al. (2018)—the deeper the color, the higher the possibility of old ice. The orange shades show the old-ice sites suggested by Van Liefferinge et al. (2018). (b) Reliability index ($\sigma_R$) map in the DF region, the reliability of the model output decreases with decreasing reliability index ($\sigma_R$) increasing.

could potentially be found (marked as dark red-dashed ellipses): the first one is a large subglacial mountain range located within a ~100 km radius around the DF drill site; the second one is ~ 160 km to the west of the DF drill site, connected with the first site; the third one is ~ 240 km to the west of the DF drill site and separated from the first two; the fourth one is ~ 260 km to the south of the DF drill site. These fourth potential candidate areas are all situated in regions with ice thickness of 2200–3000 m, where the ice is not too thick, which would result in basal melting, but still thick enough to potentially contain a long-term and sufficiently resolved ice-core record. Moreover, these sites, especially the first one close to DF, appear to be distributed over high plateaus. This could imply that the ice column here is potentially less disturbed and includes horizons of higher lateral continuity.

### 3.3 Basal thermal states

Basal conditions are crucial criteria for the presence of old ice, because any melting causes ice loss in the lowermost part of the ice column, which severely limits the age of the basal ice (Fischer et al., 2013). From our model we also obtain the basal conditions, including melt rate or stagnant-ice thickness (Fig. 6a). According to our results, basal melting prevails over frozen conditions with the formation of stagnant ice in the survey area. Modeled basal melt rates vary from 0 to 8.39 mm a$^{-1}$. Melting is significant $\sim$ 200 km south-west and $\sim$ 150 km south-east of DF, where we observe ice thicker than 3000 m, i.e. ice thick enough for the temperature to reach the pressure melting point. The basal melt rate at the DF drill site is interpolated as 0.11$\pm$0.37 mm a$^{-1}$.

Stagnant ice has a thickness range of 0–400 m. Two clusters of stagnant ice are distributed $\sim$ 60 km south-west (immediately north of NDF) and $\sim$ 180 km west of DF (in the second old-ice candidate site). The thickness of stagnant ice is modeled as 206.5 m at NDF. Our results show that melt rates are generally higher in subglacial basins and lower (or even frozen conditions) in subglacial mountainous terrain.

### 3.4 Accumulation rate

Accumulation rate is another important factor for the age distribution. We show the temporally averaged (over 720 ka) accumulation rates in the DF region from our model results in Fig. 6b. They vary from 0.015 to 0.038 m a$^{-1}$ ice equivalent. At the DF drill site, the accumulation rate spatially interpolated between the radar lines is 0.022 m a$^{-1}$. In the larger DF region, it shows a west–east decreasing gradient. In the Supplement Fig. S1 we also show the shape factor map in the DF region obtained from the model.

## 4 Discussions

### 4.1 Age of ice: comparison with previous studies

There is signifant uncertainty in the basal age of ice over the entire DF region. We relate this phenomenon to the fact that the number and depth of IRHs used as constraints for the model are limited by their traceability in our radar data set. IRH constraints help to determine the shape of the thinning function ($p$ factor), therefore, using more IRHs gives a more accurate $p$ value. The thinning function is almost linear in the upper section of the ice sheet and then becomes non-linear in the deepest part. Since the IRHs we have traced are located in the top two thirds of the total ice thickness, we cannot constrain the model well in the lower third. However, this lower section has the largest impact on the $p$ value. For comparison, in the Dome C region, the age uncertainty of each IRH is similar to that in the DF region and Chung et al. (2023) adapted the same model approach, but the modeled age uncertainty of basal ice is much smaller in the Dome C region. This is likely due to more IRH constraints covering a larger portion of the ice sheet thickness in the DC region. We consider this comparison important, as the same approach applied in different regions and/or to different radar data set can yield a considerably different uncertainty.

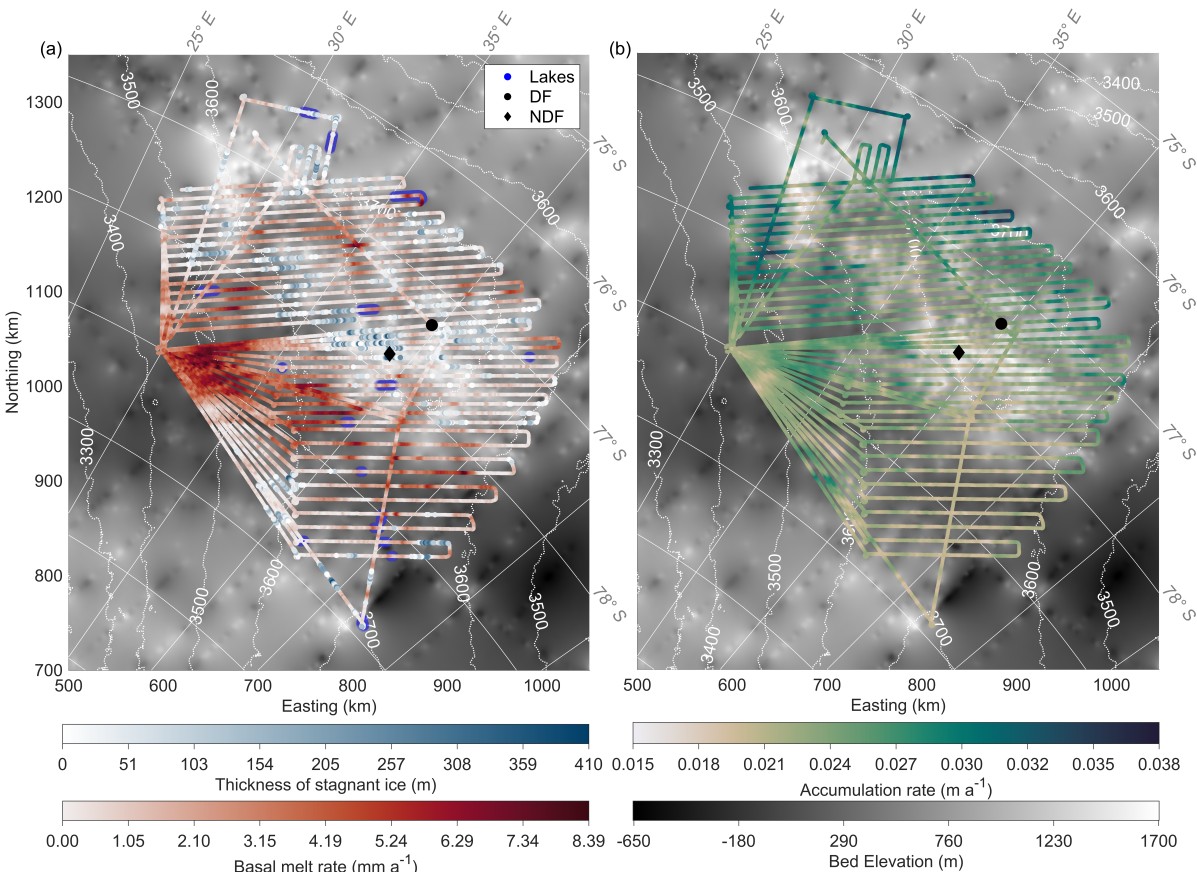

**Figure 6.** (a) Modeled stagnant-ice thickness and basal melt rate along the profiles of the radar survey: blue represents stagnant ice thickness and red represents the melt rate. Dark blue lines are subglacial lakes deduced from basal reflectivity in radargrams by Karlsson et al. (2018). (b) Modeled averaged accumulation rate in ice equivalent along the profiles of the radar survey.

Several previous studies have already investigated the potential age of basal ice either at the DF drill site or its surrounding region. At the DF drill site, Parrenin et al. (2007) proposed that ice more than a million years old could exist near the ice–bed interface, according to the results of their 1-D ice flow model. Hondoh et al. (2002) deduced chronologies of the DF ice core based on the correlation between the local metronomic signal (Milankovitch components of the past surface temperature oscillations) and the isotope record. They then extrapolated this timescale to 3050 m depth using a simplified ice-flow model. Their result suggested that age may reach 2000 ka at about 3000 m depth at the DF drill site. These two results correspond approximately to the range of our inferred bottom age of 1347.2±503.1 ka at the ice–bed interface ($\sim$ 3050.5 m).

Obase et al. (2023) used a 1-D ice flow model, which computes the temporal evolution of the vertical age and temperature profiles. They also extended their modeling results along a DF–NDF radar transect from DF to NDF, where the basal ice has a tendency to be older. They used ground-based radar data from the JARE59 survey (2017–2018) which covered an area of approximately 120 km × 100 km with a dense grid of radar lines. The data was collected using an incoherent pulse-modulated

VHF radar sounder with a peak transmission power of 1 kW, transmitter pulse widths of 60 ns and 250 ns, which corresponds
to a pulse-limited vertical resolution of 5 m and 21 m, respectively. In addition, their model is transient, therefore age and
temperature both change with time. It estimates the age through the vertical advection equation and uses the GHF as the basal
boundary condition, while we use a 1-D steady model, which calculates the age through an analytical thinning function and
excludes all thermal modeling by introducing a mechanical ice thickness.

Despite the differences in the radar data characteristics of the JARE59 and AWI surveys and slightly different models, the
results are reasonably consistent. We estimate the age of the basal ice at DF to be 1347.2±503.1 ka, while Obase et al. (2023)
extrapolated it as a range of 400–1000 ka with a GHF in the range of 60–52 mW m$^{-2}$, respectively, in their Fig. 6a. In addition,
our results confirm that the age of the basal ice is getting older from DF to NDF.

The two previous studies by Karlsson et al. (2018) and Van Liefferinge et al. (2018) are based on a thermodynamic model,
considering regions with an surface ice flow velocity smaller than 1 m a$^{-1}$. Their main constraint for the presence of old ice
is that the GHF is not sufficiently large to cause temperate conditions at the base, and thus melting. Another criterion is ice
thicker than 2000 m and 2500 m, respectively. They suggested several potential areas holding old ice, which are displayed by
semi-transparent pink and orange shades in Fig. 5, respectively. Our approach, in contrast, is solely based on the observed age–
depth distribution, which is then extrapolated to larger depth by using observed accumulation rates and making assumptions
about the thinning function.

The model we are using does not take into account the thermodynamics at all, thus it is more independent of GHF estimates.
However, the sites with potentially "old ice" suggested by the above mentioned two different approaches show a considerable
correspondence in some places with our results, especially at the first candidate (a large subglacial mountain range located
immediately around the DF drill site). We consider that the two main underlying reason for this consistency is the use of ice
thickness in both models, which has implied the important impact of ice thickness on the age distribution of the ice, as well as
the validity of the approximations regarding the thinning function in our approach.

## 4.2 Basal thermal state and accumulation rate: comparison with previous studies

A spatial comparison between our result and subglacial lakes identified previously by Karlsson et al. (2018) in Fig. 6a shows
that all 16 lakes are located in regions where we obtain basal melting, and in 11 lakes we can observe significant melting.

The basal melt rate is a parameter impacted by the spatial distribution of GHF, which is a regional parameter and also
can show strong variations on the local scale, depending on topography (Colgan et al., 2021). By averaging the local GHF
variations, we calculate regionally melt rates in different areas around the DF drill from our results for the comparison with
previous basal melt rates at DF. In our results, the mean basal melt rates increase with the distance from the DF site (i.e., in a
larger region).

Using a 1D ice flow model at the DF site, Parrenin et al. (2007) suggested that the basal melt rate is < 0.2 mm a$^{-1}$ with a
probability of 90 %. Seddik et al. (2011) deduced a basal melt rate of ∼ 0.35 mm a$^{-1}$ assuming a GHF of 60 mW m$^{-2}$. Obase
et al. (2023) suggested that there is no melting at DF for a GHF < 56 mW m$^{-2}$ and the melt rate rises to ∼ 0.4 mm a when GHF
equals 58 mW m$^{-2}$. These three results all agree with our mean basal melt rate of 0.16±0.37 mm a$^{-1}$ within 5 km around DF.

Obase et al. (2023) also simulated a basal melt rate change from 0.6 to 1.5 mm a$^{-1}$ for a GHF increasing from 60 to 64 mW m$^{-2}$ in their Fig. 5. This corresponds to our averaged basal melt rate of 1.36±0.69 mW m$^{-2}$ within 200 km around the DF site.

Talalay et al. (2020) estimated a basal melt rate of 2.5±0.5 mm a$^{-1}$ at DF based on the temperature profile measured in the ice-core borehole and an analytical solution to infer the vertical velocity. This value is consistent with our mean basal melt rate of 1.67±0.76 mm a$^{-1}$ in the entire DF region. However, this value is very different from our estimate at the DF drill site and—despite potential shortcomings in their approach—probably closer to reality. In Section 4.3.3 we discuss the possible overestimation of the basal ice age due to an inflection point at the bottom of the timescale, which would mean that we underestimate the basal melting. Figure S2 shows the mean value and standard deviation of the basal melt rates within different distances of the DF drill.

In the larger DF region, model-derived stagnant ice is only present along 8 % of the radar profiles, and has an average thickness of 95.6 m. The distribution of the stagnant ice implies that the region immediately north of NDF is the area most likely to have a cold bed which could hold old ice in the DF region. Our companion paper shows that in the DC and LDC region, the basal thermal states are very different. Stagnant ice prevails over melting in the DC area and it dominates the LDC region, with a thickness of up to 250 m (Chung et al., 2023). The relatively warm basal thermal condition in the DF region make it less likely that old ice exists. Complementary to our model results, other studies have found some evidence of stagnant ice in radargrams as notable events, e.g., no continuous or coherent reflecting horizons (Lilien et al., 2021) or diffuse scattering (Cavitte, 2017) in radar detection range. However, in the radar data set we use, there is an echo free zone (EFZ) above the bed, with a thickness of several hundreds of meters. There are various possible causes for an EFZ, e.g., sensitivity of the radar system, deformation, folding or recrystallization of ice (Drews et al., 2009; Franke et al., 2023). We consider that the EFZ in our data set is most likely caused by the performance of the radar system, as in the same region more modern systems can detect somewhat deeper, more coherent horizons (Rodriguez-Morales et al., 2020). Therefore, we could not observe any unambiguous evidence of stagnant ice in our radargrams. Fujita et al. (2012) found no features in their radar observations that could be interpreted as evidence of the refreezing basal water. They attributed this to the relatively smaller variations in bedrock topography and thus ice thickness in the DF area compared to other regions in Antarctica, where basal freeze-on was proposed. Basal melt water would thus favorably drain downstream in the DF area than to follow paths which would enable local freeze-on locally.

In the DF region, Fujita et al. (2011) showed an map of accumulation rate with a decreasing trend from 76° S to 78° S, which is consistent with the distribution of accumulation rate in our result.

## 4.3 Reliability and sensitivity study of the 1-D model

### 4.3.1 Reliability of the model

Our 1-D model does not consider horizontal advection, which, although low near an ice divide, exists away from the divide. In these places, the reliability of the model is lower. We show the reliability index $\sigma_R$ (described in Section 2.4) in the DF

**Table 1.** Spatial standard deviation of age of basal ice at different distances from DF and NDF.

| Distance (km) | Standard deviation (ka) | |
| | Around DF | Around NDF |
| --- | --- | --- |
| < 5 | 392.7 | 50.9 |
| 5–15 | 487.0 | 369.3 |
| 15–50 | 526.5 | 529.6 |
| 50–100 | 531.2 | 545.7 |
| 100–200 | 497.7 | 517.7 |
| > 200 | 534.9 | 535.5 |

region (Fig. 5b). A smaller reliability index $\sigma_R$ represents higher reliability of the model. The reliability index $\sigma_R$ ranges from 0.1 (reliable) to 1.2 (less reliable) in the DF region. The distribution indicates a relatively higher reliability in the DF region compared to that in the DC region (0–2) (Chung et al., 2023). The reliability of the model in the DF region could be overestimated because of the limited number and depth of IRHs.

To evaluate the reliability of the model results in ice deeper than the available IRHs, we also determine the spatial deviation of the age of basal ice at different distances from DF and NDF as a function of normalized ice thickness. The underlying assumption is that the age–depth function should be rather similar for the same (normalized) ice thickness within a region for small flow velocities and where the overall ice dynamic behaviour (e.g. prevailing divide or flank flow regime) is comparable. The spread of the distribution of age of deep ice, shown in Table 1, generally increases with distance from DF and NDF, except

between 100 and 200 km distance from both sites and >200 km from NDF. The distribution of the age–normalized depth is illustrated in Fig. S3 in the supplement.

We interpret the larger spread to reflect the increasing transition from a dome-flow to a flank-flow regime. Within the region of clear characteristic of divide flow, i.e. a relatively smaller spread, implies that it is reasonable to apply a 1-D model. Our approach is comparable to the method mentioned above used by Karlsson et al. (2018) and Van Liefferinge et al. (2018) to

investigate the age only in areas with an ice flow velocity $< 1$ m a$^{-1}$ (in ice equivalent). The standard deviation of the age distribution 5 km around NDF (50.9 ka) is much smaller than that around DF (392.7 ka). This could tentatively be interpreted as flow characteristics near NDF representing more homogeneous divide flow than those at DF.

### 4.3.2   Sensitivity study

We next discuss sensitivity studies with which we investigate how different data inputs and constraints affect the model and

how the reliability of the model could be improved.

The thinning function and the normalized age–depth scale have a stronger gradient in deeper ice than at shallower depths. Therefore the deepest horizon as well as the underlying age–depth scale may have an effect on our modeling results, including

shape factor $p$, accumulation rate $\dot{a}$, mechanical ice thickness $H_m$, and age of basal ice $\chi_b$. To investigate these effects, we perform two sensitivity experiments for the profile 20170240.

Our first run corresponds to the standard model run (STD) which we have been discussing so far, i.e. it uses six or seven traced IRHs and DFO2006+AICC2012 as the timescale. The timescale provides the temporal variations of the accumulation rate at DF and allows us to date the IRHs.

The second model run (RUN II) investigates the impact of using a different number of traced IRHs to constrain the model. In order to give a better constraint, an extra deeper discontinuous eighth horizon, EH8 with an age of 232.7 ka, was traced (Fig. 2). As this IRH is discontinuous in the study region, it could not be used reliably on all other radar profiles, but still provides a useful addition on those profiles where it is present.

In the third run (RUN III), we analyze how different timescales influence the modeling results. We use IRHs H1–H7 from the standard run as constraints, but replace DFO2006+AICC2012 with DFGT-2006. Parrenin et al. (2007) reconstructed the age from the first DF ice core using a 1-D flow model, referred to as DEGT-2006. Below 2503 m (the depth of the first deep ice core), the temporal variations of accumulation rate could not be as reliably reconstructed in DFGT-2006 as for other ice cores, as it was derived from marine cores. Therefore, to increase reliability, we use the temporal variations of accumulation rate below 2503 m from timescale DFO2006+AICC2012 as a replacement.

In order to quantify the difference between model results from different runs, we provide statistic values of relative percentage difference of shape factor $\Delta p$, accumulation rate $\Delta \dot{a}$, mechanical ice thickness $\Delta H_m$, and age of basal ice $\Delta \chi_b$ along the profile 20170240 between STD and RUN II, STD and RUN III in Table 2, respectively. In the following paragraphs we discuss the results. For extended illustration we refer to the supplement, where we provide and analyse the model results for all three runs Fig. S4 and relative percentage difference of model results between STD and RUN II/RUN III in Fig. S5.

The outcome of RUN II shows that the age of the basal ice and the shape factor are severely affected by an extra IRH (mean $\Delta p = 12.55\,\%$ and mean $\Delta \chi_b = 14.60\,\%$). In some regions, EH8 has a somewhat different shape to the upper seven IRHs and thus changes the inferred shape factor and thinning function of each modeled point, which leads to significant change in the age of basal ice. Between STD and RUN III, the mean relative difference of age of the basal ice and the shape factor are also large (10.43 % and 9.07 %). They thus prove the importance of using the most reliable ice-core timescale. The standard deviation of $\Delta p$ and $\Delta \chi_b$ are significant (6.09 % and 10.63 %), which could be related to the changes in subglacial topography.

The relative percentage difference $\Delta \dot{a}$ (absolute values) of STD minus RUN II and RUN III has a mean value of 0.83 % and 3.20 %, respectively, which implies the accumulation rate is almost unaffected by adding an extra IRH and affected more by the timescale. The standard deviation of $\Delta \dot{a}$ between STD and RUN III is low (0.18 %), which proves that the relative difference seems less variable along the profile. We therefore suggest that using different temporal variations of accumulation rates at DF could be the main reason for the difference of the modeled accumulation rate between STD and RUN III. This is, however, not surprising, as accumulation has a larger influence on isochrones near the surface and a smaller influence on the ones at larger depth (Sutter et al., 2021).

**Table 2.** Mean value and standard deviation of relative percentage difference between model runs for the profile 20170240.

| | $\Delta\chi_b(\%)$ | | $\Delta p(\%)$ | | $\Delta\dot{a}(\%)$ | | $\Delta H_m(\%)$ | |
|---|---|---|---|---|---|---|---|---|
| | Mean | Std. dev. | Mean | Std. dev. | Mean | Std. dev. | Mean | Std. dev. |
| STD–RUN II | 14.60 | 14.70 | 12.55 | 10.07 | 0.83 | 0.71 | 4.35 | 3.34 |
| STD–RUN III | 10.43 | 10.63 | 9.07 | 6.09 | 3.20 | 0.18 | 3.15 | 0.60 |

Mean values of the relative change in the mechanical ice thickness $\Delta H_m$ imply that both, the number of IRHs (4.35 %) and change of agescale (3.15 %) have a comparatively small impact on the deduced mechanical ice thickness. This implies that the mechanical ice thickness obtained from our model is relatively robust compared to other quantities.

### 4.3.3 Comparison of the age–depth scales

Comparing the age–depth distribution at the DF drill site of the three model runs (Fig.7), we find that at depths larger than $\sim 2500$ m, the three runs have very similar age–depth scales. The differences between STD and RUN II and RUN III are much larger below a depth of 2500 m, where STD has an age of basal ice of 1347.2±503.1 ka. RUN II with an extra horizon results in an age of the basal ice of 1958.0±726.1 ka, while RUN III uses the input from a different timescale and obtains an age of 1933.7±769.3 ka at the basal ice. This comparison shows that both, the number of IRHs and the agescale, have a significant

influence on age of the basal ice. Since RUN III uses the extrapolated timescale (DFGT2006), not the timescale of the second DF deep ice core determined by ice-core analysis (DFO2006+AICC2012), we will not consider it further in the discussion.

If we only focus on the age of basal ice, it seems that both modeled ages (STD and RUN II) deviate from their timescale DFO2006+AICC2012. The modeled age of STD even seems more reasonable than the modeled age of RUN II with one less IRH, although it is important to note that there is huge uncertainty for Run II and STD. However, since we cannot simply and

395 independently assess the quality of the model results based only on the age of basal ice, we will next analyse the complete age–depth profiles and discuss the age–depth distribution of each RUN.

Comparing the modeled ages of STD and RUN II with their timescale (DFO2006+AICC2012), we find that above $\sim 2350$ m, both modeling results have good agreement with the timescale. From $\sim 2350$ m to $\sim 2745$ m, only the modeled age of the RUN II agrees with the timescale. The STD modeled age is numerically smaller than the age from the timescale, the difference

between them increases with the depth. This finding shows the significant impact of the extra IRH EH8 in RUN II: with one more IRH which is 257 m deeper and 63.6 ka older than the one above, the modeled age stays comparatively accurate for a further $\sim 395$ m in depth.

RUN II has a reasonable performance down to a depth of $\sim 2745$ m, where the age is modeled as 536.4 ka BP. This depth is $\sim 300$ m above the bed, which is exactly the depth of the inflection point in the timescale DFO2006+AICC2012. Below this

depth, the age–depth profile of the model keeps following the exponential distribution as a model assumption, but the timescale of ice core shows a curvature reversal. Thus, the modeled age gradient is steeper in the same depth range, which leads to the

large overestimation of the age of basal ice by a factor of two in this case. Fig. 4d depicts the spatial distribution of age of ice at 300 m above the bed. It provides relatively accurate age values while excluding the lowest part of the ice. The age has a range of 153 ka–989 ka, and implies that there is a small area for old ice at this depth $\sim$ 200 km south–east to DF.

Obase et al. (2023) shows this inflection at the same depth ($\sim$ 300 m above the bedrock) in their Fig. 6a, and it also caused a much older modeled age compared to the observation. Such an inflection point in the age–depth scale obviously indicates that the underlying analytical assumption for our model approach is less valid below its depth of occurrence.

A similar phenomenon was also observed in our companion study with the same model approach (Chung et al., 2023) at EDC, though much older IRHs (up to 476.4 ka BP) were dated there. They pointed out that the modeled age at the deepest
dated point for the EDC drill site was around 100-200 kyr older than would be expected from the AICC2012 age–depth profile. The reason could be that the profile of the timescale AICC2012 determined by ice-core analysis does not follow an exponential profile in the lower 200 m of dated ice. Since the timescale of EDC does not change as drastically as that of DF ice core near the bedrock, the overestimation of the modeled age of basal ice at EDC is not as significant as that at DF. For illustrative purposes we also show the model derived age–depth scale and AICC2012 timescale at EDC in Supplement Fig S6.

To solve this overestimation problem in the case of DF, only more continuous isochrones below the inflection point, i.e., the lowest 300 m, would provide better constraint for the model. This is not possible with our radar data set and is also unlikely to be easily achievable in other data sets (e.g. Tsutaki et al., 2022).

The logging of the DF borehole and the drilling process indicated melting at the base of DF (Motoyama et al., 2021), which could be one reason for the inflection point in the timescale of the ice core. This would imply that the significant overestimation
likely occurs in areas with basal melting. Given that there is only one deep ice core in the DF region, we lack an additional timescale extending towards the bedrock to prove our hypothesis. Whether the inflection point in the age–depth profile is a general feature in the DF region is still an open question.

Overall, we find that our model works quite well in the upper two thirds of the ice column according to the calculated reliability index (the standard deviation of the age difference between observation and model results). It also seems appropriate
in the deeper part of the ice column at DF, down to the depth of the inflection point of the timescale. Since we have found that areas of basal melting prevail over those with stagnant ice in the DF region, it is possible that overestimation of age occurs in the deep ice at various places, as what was already observed at DF. Taking this into consideration, we emphasize the importance of considering the basal thermal state for locating old ice, i.e., more attention should be paid to areas indicating the presence of stagnant ice.

## 4.4  Limitations of radar system and model

In the following we will discuss the current limitations from the perspectives of the radar system and the model, respectively, in order to point out potential improvements for future approaches.

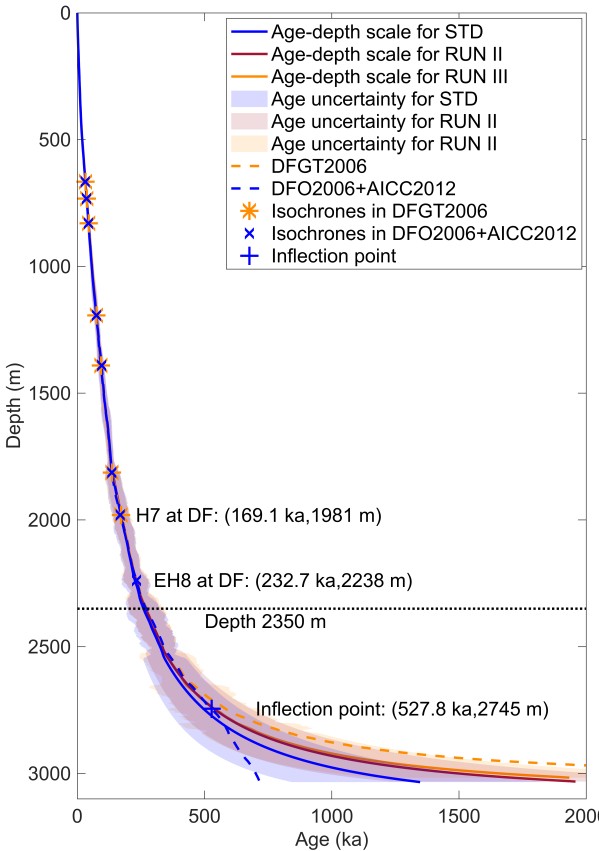

**Figure 7.** Comparison of age–depth scales of three models (solid lines), their uncertainties (shades) and two timescales from the DF ice core (dashed line). Note that the uncertainties of RUN II and III are similar, so their shades are overlapped mostly. The asterisks and crosses show the age and depth of IRHs for the DFGT2006 and DFO2006+AICC2012 timescale, respectively. The plus sign shows the inflection point of the timescale DFO2006+AICC2012. Labels at the distribution indicate horizon, age and depth.

### 4.4.1   Radar system limitations

According to the sensitivity study and comparison of age–depth scales, the shape of IRHs (i.e., the accuracy of tracing) and
the depth of IRHs have a significant impact on the modeled age of the ice. However, compared to modern state-of-the-art radar systems, the data collected by the AWI RES system with a pulse width of 50 m, leads to marked higher uncertainties during manual IRH tracing and thus lower reliability of the model results. The lower resolution and SNR also limits the number of traceable IRHs, which in turn increases the age uncertainty in the bottom part of ice. In addition, although we traced the deepest continuous horizon at 169.1 ka in this study, the lowermost third of the ice column is still not dated yet. The lack
of clear coherent return signals in lowermost part likely originates not only from the physical properties of the ice but also from the limitations of the radar system. Rodriguez-Morales et al. (2020) investigated the comparison of data collected with the ground-based CReSIS' UWB radar, Japanese National Institute of Polar Research (NIPR) radar and the AWI RES system

(600-ns burst) along semi-coincident survey trajectories in the DF region. Their results implied that at the same depth modern systems would provide not only a higher resolution, but most likely also a deeper detection of continuous IRHs (Rodriguez-Morales et al., 2020). In addition, as we pointed out in Section 4.2, the EFZ visible in our radargrams is most likely caused by the radar system itself, which disables us to find the correspondence events of stagnant ice in the data.

Our sensitivity study also showed the correspondence between the age of basal ice and ice thickness, as a crucial input in the ice-flow model. Accurate ice thickness can improve the reliability of the modeling results. Our radar data were collected with an incoherent burst radar system, which means hyperbolic effects in signals are strong and affect the accuracy of subglacial topography. According to Tsutaki et al. (2022), the average difference between ice thickness observed from the JARE radar system (with high-gain and high-directivity antennae) and the AWI RES system is −8 m, and the standard deviation is 108 m. The high standard deviation implies the details of the bed topography observed by two radar systems could be significantly different, which may cause the misalignment in modeling results.

Overall, the radar system we use in this study limits the number, depth and accuracy of the IRHs traced, the possibility of observing the basal unit and the resolution of bed topography observed in the radargrams, which all affect the modeling results. In contrast, despite these shortcomings the simple and light-weight system enabled a long range of the aircraft from a high-altitude field camp to cover a large region around DF. Analysis of ground-based radar observations from more sensitive radar systems with higher vertical as well as horizontal resolution in sub-regions of our larger DF area, as were already acquired in the past, will thus complement the large-scale results from our study with more accurate detailed insights.

### 4.4.2 Modelling limitations

Our model does not consider horizontal advection and assumes that the basal sliding ratio is negligible, which are proper assumptions at DF. To improve the reliability of the model results in regions further away from DF, a basal sliding term could be added. However, this would make it more difficult to infer the mechanical ice thickness, the velocity shape exponent and the sliding ratio at the same time. Furthermore, although 3-D full Stokes models can lift restrictions, they still come along with new challenges, including heavy computation time, more complicated boundary conditions and conjunction between 3-D model and age observations. For the time being, a model of intermediate complexity operating along flow lines or 2.5D approaches might provide useful results nevertheless (Gerber et al., 2023)

### 5 Conclusions

We utilized a 1D ice-flow model to reconstruct the age field and analyse the basal thermal states in the DF region. The model is constrained by traced internal horizons observed in airborne radar data, which are dated by transferring ages from the DF ice-core timescales.

According to the modeled age of the basal ice, we identify four potential candidate areas for old ice in the DF region: a subglacial mountainous target located around the DF drill site with a radius of $\sim$ 100 km, $\sim$ 160 km to the west of the DF drill, $\sim$ 240 km to the west of the DF drill and $\sim$ 260 km to the south of DF drill. The first candidate deserves most attention

since it has a good correspondence with the previous old ice predictions obtained by a very different model approach (Karlsson et al., 2018; Van Liefferinge et al., 2018). At the DF drill site, the modeled age of the basal ice is 1347.2 ka$\pm$503.1 ka . At NDF the maximum age is extrapolated as 1472.1 ka$\pm$509.0 ka at a depth of 2080.7 m. The age of basal ice has a considerable uncertainty due to limitations in the number and depth of our IRHs, which could be mitigated by using radar data set with higher resolution, higher sensitivity and thus better traceability. Deployment of state-of-the-art radar systems might decrease

this limitation and lead to improved model performance.

The modeled basal thermal state implies that melting is more common than stagnant ice in the DF region. Modeled basal melt rates vary from 0 to 8.39 mm a$^{-1}$. Melting is significant $\sim$ 200 km south–west and 150 km south–east of the DF drill site. At the DF drill site our model produces a melt rate of 0.11$\pm$0.37 mm a$^{-1}$, which corresponds to earlier estimates. Stagnant ice is mainly present immediately north of NDF and $\sim$ 180 km west of the DF drill site. It occupies only 8 % of the radar profiles

with an average thickness of 95.6 m. The region close to NDF has the most favorable conditions for a cold bed holding old ice. The thickness of stagnant ice is modeled as 206.5 m at NDF.

We obtain an average accumulation rate over the past 720 ka of 0.015–0.038 m a$^{-1}$ ice equivalent in the DF region and 0.022 m a$^{-1}$ ice equivalent at the DF drill site.

In our sensitive study we demonstrated that an extra IRH at deeper depth and/or using a different timescale significantly affect

the model results. This underlines the importance of using IRHs traced as deep as possible and to use the most trustworthy timescale to get reliable model results. The radar system we use in the study limits the number, depth and accuracy of the IRHs traced, the possibility of observing the basal unit and the resolution of bed topography observed in the radargrams, which all affect the modeling results. Using ground-based observations from improved radar systems with higher vertical as well as horizontal resolution in sub-regions of the larger DF area, as were already acquired in the past, will complement the large-scale

results from our study. Our model approach is based on assumptions, like ignoring horizontal advection and basal sliding. A 3D model would partly lift these simplifications and potentially improve the reliability of model results, but would also increase the demand for computing resources and boundary conditions.

We observe an inflection point at the depth of $\sim$ 300 m above the bed in the experimental timescale of the DF ice core, which we consider to be caused by more complex flow related to basal melting. This shows the inability of our model to

505 capture complex thinning phenomena below that depth and thus causes an overestimation of the age in the lowermost ice. Thus, we recommend to consider the modeled age of the ice shallower than 300 m above the bed (roughly 10% of the ice thickness) for decision making. At the same time, more attention should be paid to the basal thermal state, which is likely a hidden factor implying the accuracy of the modeled age. Considering the age of ice and basal thermal state together, we suggest that the area immediately north of NDF could be a potential old ice drill site in the DF region.

*Code availability.* The model code is available from Github. https://github.com/ailsachung/IsoInv1D

*Data availability.* The IRH data are available from the PANGAEA repository https://doi.org/10.1594/PANGAEA.958462 (Wang et al., 2023). The bed elevation and surface elevation data set collected by NASA Making Earth System Data Records for Use in Research Environments (MEaSUREs) program (Morlighem et al., 2017, 2020) are available on https://nsidc.org/data/nsidc-0756/versions/3. We used elevation data from the Antarctic Mapping Tools in MATLAB from https://de.mathworks.com/matlabcentral/fileexchange/47638-antarctic-mapping-tools (Greene et al., 2017). The ice thickness derived from radar data are published on https://doi.org/10.1594/PANGAEA.920234 (Eisen et al., 2020).

*Author contributions.* OE coordinated the BE–OI project and designed this study. ZW carried out experiments and wrote the manuscript with input from all co-authors. ZW and DS processed the radar data and traced the horizons. FP developed the model. AC improved and adapted the model to the conditions at Dome Fuji and performed the calculation. JF drilled the B53 ice core and provided the depth–density profile of the ice core. All coauthors read and commented on the manuscript.

*Competing interests.* O.E. is an editor of The Cryosphere. The authors declare no other competing interests.

*Acknowledgements.* This publication was generated in the frame of Beyond EPICA-Oldest Ice (BE-OI). The project has received funding from the European Union's Horizon 2020 research and innovation program under grant agreement No. 730258 (BE-OI CSA). It has received funding from the Swiss State Secretariate for Education, Research and Innovation (SERI) under contract number 16.0144. It is furthermore supported by national partners and funding agencies in Belgium, Denmark, France, Germany, Italy, Norway, Sweden, Switzerland, the Netherlands, and the United Kingdom. Logistic support is mainly provided by AWI, BAS, ENEA and IPEV. The opinions expressed and arguments employed herein do not necessarily reflect the official views of the European Union funding agency, the Swiss Government, or other national funding bodies. We thank the logistics field team and flight crew for support during the expedition. We thank Emerson E&P Software, Emerson Automation Solutions, for providing licenses in the scope of the Emerson Academic Program. We thank Brice Van Liefferinge for providing the locations of the promising old ice in the DF region in his previous study. We thank Nanna B. Karlsson, who offered the detailed locations of the subglacial lakes and the old ice candidates in her previous study. We thank Kenji Kawamura for the insightful discussions on the DFO2006+AICC2012 timescale of the Dome Fuji ice core. Zhuo Wang is funded by China Scholarship Council (No. 202106170102). Thank goes to Professor Zhaofa Zeng for supporting her in getting the grant and jointly studying abroad as her supervisor in Jilin University. Ailsa Chung is funded through the DEEPICE project, from the European Union's Horizon 2020 research and innovation program under the Marie Sklodowska-Curie grant agreement No. 955750.

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
