# Peer review of "Mapping age and basal conditions of ice in the Dome Fuji region, Antarctica, by combining radar internal layer stratigraphy and flow modeling"

_The Cryosphere, 2023_

## Author Comment (AC2)

Response to RC 1 of 'Mapping age and basal conditions of ice in the Dome Fuji region, Antarctica, by combining radar internal layer stratigraphy and flow modeling' (tc-2023-35)

May 4, 2023

*The paper provides an assessment of the distribution of ice that meets the targets of the International Partnerships in Ice Core Science "Oldest Ice Challenge" using a kinematic approach inverting observed englacial isochrons dated using an ice core pinning point for the deep age structure of the ice sheet. This 1-D approach (specifically IsoInv) has been primarily used for the Dome C region, and stands in contrast to 1D thermodynamic approaches that balance both mass and heat through latent heat (eg van Liefferinge et al., 2014, 2018), and 3D full stokes modeling. The approaches are complementary, with their strengths and weaknesses - thermodynamic approaches have to deal with the tremendous uncertainty in the value of geothermal heat flux, while attaining the deep dated isochrons needed for IsoInv is fraught and the computational cost and rheological and boundary condition uncertainties for 3D approaches limit their usefulness.*
*This paper represents an overall readable contribution to the literature on old ice distribution modeling. I don't have fundamental problems with the conclusion. Aspects of the presentation and discussion would benefit from being made clearer.*

We thank Anonymous Referee #1 for taking the time and effort to provide us with the helpful comments. We believe the manuscript will be improved by following these comments. We have discussed on all the comments from referee and will respond to all of them in order. For clear tracking of changes, individual issues raised by the referees are referred to as (**Revision I**) below, where "I" is the number of the comment in the attached table.

*Major points:*
*Gray literature: The paper does lean on preprints somewhat. The study by Chung et al., submitted is obviously highly coupled with many of the same authors - one would anticipate that the methods this paper refers to will not change greatly, but there is a risk. A larger issue is the pointers to the preprint by Obase et al. 2022, looking at complementary thermodynamic modeling where the reviews indicate that there was an error in the calculation of the basal thermal gradient. Presumably this is addressed in the final accepted (but not yet published) version of that paper. While the Obase paper is obviously very complementary, the Wang paper should either drop the explicit comparisons, or at least provide some caveats on that comparison.*

We thank the reviewer for pointing out the gray literatures.
- The study by Chung et al. (2023, in review) is our companion manuscript, the IsoInv model was used in both studies, and Chung et al. (2023, in review) gave a detailed description of the model in their manuscript submitted. In our manuscript, we cited their research for either directing people who is more interested in the model towards their study (Line 69 and Line 162) or compare the basal thermal state (Line 257) and model reliability in the DF and DC region (Line 267). Nevertheless, regarding the gray literature issue, we will add further details of the model, i.e., the optimization algorithm in the Method section. Therefore we can make this part more independent from our companion research. (**Revision 1**)
- As the reviewer has pointed out, the Obase et al. (2022, in review) paper is very complementary by using a 1D temperature and age model which takes geothermal

heat flux into account. We will keep the comparison in our manuscript for the time being and point out the caveats **(Revision 2)**. We will rephrase the comparison in our manuscript depending on the final state of the Obase et al. (2022, in review) manuscript in case our manuscript is accepted before theirs is finally published.

*__Figures:__ The maps are hard to read, especially at printed scale.  I recommend removing the greyscale DEM bed elevation background and using some well spaced contours instead in a different color than the surface contours.  Increasing the plotted point size of the data point may also help make the ideas conveyed more visible.  The authors discuss specific candidate old ice sites in the text - they should indicate them on at least one of the maps.  The indicators for DF and NDF are hard to make out at printed scale. More specific points are below.*

Thanks for the suggestions on figures. We follow the suggestions and replot the Fig 1, 4, 5, 6. **(Revision 3, 4, 5, 6)**

- We replace the gray scale bed elevation background with colored contours with different line style in Figure 4a. We haven't adapted this change to other figures yet since we found the corrected one seems less readable. We put the new Figure 4 here as the example, the left one use the contour to replace the gray background. It is very difficult to be concise and at the same time be very inclusive and take into account the needs of colour blind people. We tried to to that but at a certain point a limit is reached. Here we would kindly ask the reviewer for guidance.

[Figure]

- We will increase the size of data points.
- We will add four ellipses to Fig. 5a (Modelled age of basal ice) to clarify the locations of old ice candidates. **(Revision 5)**
- We will increase the size of markers of DF and NDF in the figures.

*Flank flow versus divide flow: 1-D models are really only appropriate where ice velocities are very low, as the authors point out, but the case they make for Dome F could be made better. They use the statistical spread of basal ice ages as a function of distance from key Dome points as an indicator for flank flow, but don't make their logic clear on why that should be the case. Isn't this statistical trend just a fractal distribution as you cover more and more area with expanding range from the dome? A map of ice flow velocity, or an indication of ice velocity on the maps would be helpful.*

We agree that 1D is appropriate where ice velocities are low. But how low? That's what we would like to illustrate with this analysis. As the distribution are normalized they should in principle be independent of ice depth and flank flow and thus overlap more strongly than if those regimes are included, where flank flow becomes more dominant. As we consider this a useful and simple illustration of the underlying logic, which we will make more clear in the revision, we can also consider to discard this analysis if the editor or reviewer consider it too far from the main case. We will provide a map with velocities in the revision. **(Revision 7)**

*Age uncertainties of internal reflection horizons: This was confusing. How are depth uncertainty, range precision and best guess uncertainty combined? Is range precision actually calculated from using the SNR? If so, where are the results (which should be different for each IRH).*

Sorry for the confusing description. People have used different methods to evaluate the uncertainty of horizon positions. Lilien et al. (2021) used quarter-wavelength uncertainty and Cavitte et al. (2016) use a concept of range estimate precision. In our research, we follow the latter method, because we want to involve the impact from sub-resolution of different horizons.

The range estimate precision in determining reflection depth is determined by the pulse width of the radar waveform, the signal-to-noise ratio and the sub-resolution reflector fluctuations. The last term could be ignored when the reflections have " continuality of reflection amplitudes and subsequent traceability" (Cavitte et al., 2016), but this is not the case in our study.

We use radar data collected by AWI RES system with 600 ns in this study, which means 50 m vertical resolution in the radargram. The resolution is lower than that of more advanced radar systems, which causes lower subsequent traceability. Moreover, the bedrock topography that is characterized by a series of mountain ranges and valleys and wide melting distribution in the Dome Fuji region, lead to the discontinuity of isochrones at some places, especially near the bottom. These reasons mentioned above could explain that we need to consider the sub-resolution of different reflectors.

We found that the uncertainty caused by the low traceability and continuity is actually large when we traced the horizons semi-automatically, which means we need to choose where to trace the horizons in the disturbed discontinuous places with a few possible trace routes. E.g., in the radar segment shown below, the horizon in the blue frames is easy to be traced, but between the frames, the stratigraphy is harder to follow and we need to interpolate it referring to the shape and pattern of other internal layers. This is a standard approach for instance also used in marine geophysics to interpolate stratigraphic boundaries. The uncertainty of different isochrones ranges from 20 m to 50 m for different horizons. We call it best-guess uncertainty, since it is also impacted by the signal-to-noise ratio (SNR) and resolution. As it

is also larger than the range estimate precision calculated from the SNR and range resolution, we finally took it as the uncertainty of horizon positions. To conclude, the tracing process brought larger uncertainty with it mainly because of the resolution of radar system, compared to radar systems used in other research. This large uncertainty needs to be estimated for different horizons during tracing, and considered into the uncertainty part.

[Figure]

We will rephrase this part in a clearer way in revision. **(Revision 8)**

*Stagnant Ice:* *At Dome C the stagnant layer has a distinctive radar character. The authors should comment either if similar features are seen at Dome Fuji, or if the radars that have been used can even detect it.*

In the radar dataset we used, there is an echo free zone (EFZ) above the bed, with a thickness of several hundreds of meters. EFZ could be caused by a several reasons, e.g., system sensitivity of the radar system, deformation, recirculation and recrystallization of ice (Drews et al., 2009 and Franke et al., 2023). We think the EFZ in our dataset could be caused by the performance of the radar system, since in the same region more sophisticated radar systems can detect deeper signatures, as we have pointed out in our manuscript (L340), 'It implied that at the same depth modern systems would provide not only a higher resolution, but most likely also a deeper detection of continuous IRHs (Rodriguez-Morales et al., 2020).'

We will point it out in the section 4.2 when we discuss the basal thermal state and 4.3.4 when we discuss the limitation of radar system. **(Revision 9)**

*Specific points:*
*Abstract line 6:* *probably best to use 'basal unit' to replace 'basal layer' following on Lilien et al., 2021.*

Thank you, a reasonable suggestion. But since the term "basal unit" is defined in the radargram and "stagnant ice" is what we deduce from the model, we will use "stagnant ice" to replace "basal layer" here. **(Revision 10)**

*Line 26:* *replace ''feasible' with 'useful'*

Done. **(Revision 11)**

*Lines 37-39:* *The Bo 2014 paper, with the 3-D model, is more of a point to the issue discussed above – they conclude that "Hence, with the observations available now, we cannot constrain the age of the basal ice well.". The 1.5 million year where the ice is not melting from their Figure 6 is an assumption used to drive their model thermodynamics, not a result.*

Thanks for pointing out the misunderstanding. We will change Line 36-39

"In the Dome A region, Sun et al. (2014) estimated ice age around Kunlun station by applying a three-dimensional, thermomechanically coupled full-Stokes model, which indicated that in the area without basal melting the ice age at 95 % depth could be limited to 1.5 Ma."

to

"In the Dome A region, Sun et al. (2014) estimated the age of ice around Kunlun station by applying a three-dimensional, thermomechanically coupled full-Stokes model assuming different geothermal flux and fabrics. They imposed a 1.5 Myr limit to the age solver, thus they did not get the actual age of the oldest ice, but the distribution of ice potentially older than maximum run time of their model." **(Revision 12)**

*Lines 40-:* *the paragraph starting at line 40 is very long and dense and could be broken up.*

Done. We will split the paragraph to two parts, the researches of the Dome Fuji ice core and in the large Dome Fuji region. In addition, we will also add the conclusions of the previous studies follow the suggestion from another reviewer. **(Revision 13)**

*Line 56:* *"on an airborne radar surveys" should be "on airborne radar surveys" or "on an airborne radar survey".*

Done. **(Revision 14)**

*Line 89:* *what is a 'two-dimension filter'?*

It is a particular filter in the software "Echos", which we use for analysis. Since the filter could be adapted to the radar data in both trace direction and time direction, it is called a 2D filter. This filter returns a weighted running average. It is summing up amplitudes at the same travel-time, which could remove the horizontal noise. We will change
"…a low-pass filter and a two-dimension filter are…"
to
"…a low-pass filter and a running average filter are…" **(Revision 15)**.

*Line 98:* *"in all survey lines, the third IRH H3 is" change to "in all survey lines, however the third IRH H3 is"*

Thanks for the comment, we realized that we have not expressed ourselves clearly enough in the text. We will modify Line 98
"We trace 6 or 7 relatively distinct and continuous IRHs (H1 – H7) in all survey lines, the third IRH H3 is not clear and continuous enough to be traced in some profiles"
to
 "We trace 6 (H1, H2, H4 – H7) or 7 (H4 – H7) relatively distinct and continuous IRHs in the radar profiles, since the third IRH H3 is not clear and continuous enough to be traced in some profiles." **(Revision 16)**

*Line 128:* *"The estimate of the range precision is always higher than the resolution" - the numerical value for precision should be smaller than the resolution for a well behaved echo waveform; I would not use the term "larger" to mean "better". maybe finer verse courser?*

Thanks, we will change
"The estimate of the range precision is always higher than the resolution"

to
"The estimate of the range precision is always numerically smaller than the vertical resolution". **(Revision 17)**

*Line 264: for lazy readers who skip the methods, I would add "We show the reliability index (described in section 2.4)"*

Done. **(Revision 18)**

*Data availability: It would be good to get the IRH data at least in a repository prior to acceptance. Technical issues with getting the radar data are more understandable, but we should as a community be moving toward getting that as well. For the ice thickness product used for the modeling, would it be more appropriate to point to the Eisen et al., 2020 (https://doi.org/10.1594/PANGAEA.920234) product for the line-based data?*

We submitted the IRH data to Pangaea before submitting the manuscript, it took some time to be published and get the registered doi. The data is now available on https://doi.org/10.1594/PANGAEA.958462.

The ice thickness product is provided by Karlsson et al., 2018 (https://doi.pangaea.de/10.1594/PANGAEA.891323). In section 2.2 Line 99, after "Ice–bed returns were picked by Karlsson et al. (2018) through semi-automatic detection routines in MATLAB" we will add "This ice thickness data is available on PANGAEA (Karlsson et al., 2018)". **(Revision 19)**

*Figure 1: While Greene et al., 2017 should be cited if AMT was used for these plots, Greene et al is not an appropriate citation for the surface elevation data. AMT provides at least 3 different surface DEMs for Antarctica, and this paper should reference the one ultimately used.*

We have used the BedMachine plugin in AMT to plot the figure. The BedMachine data has two references, one is a data product, one is a data paper. We will adjust the citations from
"… from Greene et al. (2017) and Morlighem et al. (2017, 2020) …"
to
"… from Morlighem et al. (2020) and Morlighem (2022) ..." **(Revision 20)**

*Figure 2: what is the strong line at ~250 m depth?*

This is the radar blind zone below the surface reflection, due to saturation of the amplifier of the receive channels of the radar. The strong line is the bottom of this blind zone.

*Figure 5: gray polygons (the Van Liefferinge et al., 2018 data) on a gray scale map does not work well*

We will change the gray polygons to another color. **(Revision 5)**

*Figure 6: the patches of blue stagnant ice are nearly invisible in this rendition. It might be better to have a separate figure or indicate existence rather than thickness. The distribution with respect to lakes you have here is interesting with comparison to Dome C where we apparently have lakes under stagnant ice.*

We will increase the size of the scatters and combine the to make stagnant ice clearer **(Revision 6)**.

Sometimes there is basal melting underneath the basal unit, as the basal unit is advected from regions of thinner ice. We will consider this comment and compare the lakes distribution in Dome Fuji and Dome C. **(Revision 21)**

*Figure 7:* *it's very hard to tell what is going on with the overlapping color zones. Especially if one is color blind - STD and Run III could look identical.*

We will use different color to make figure more visible. **(Revision 22)**

**References**

Chung, A., Parrenin, F., Steinhage, D., Mulvaney, R., Martín, C., Cavitte, M. G. P., Lilien, D. A., Helm, V., Taylor, D., Gogineni, P., Ritz, C., Frezzotti, M., O'Neill, C., Miller, H., Dahl-Jensen, D., and Eisen, O. (2023): Stagnant ice and age modelling in the Dome C region, Antarctica, EGUsphere, 2023, 1–31.

Drews, R., Eisen, O., Weikusat, I., Kipfstuhl, S., Lambrecht, A., Steinhage, D., Wilhelms, F. and Miller, H. (2009): Layer disturbances and the radio-echo free zone in ice sheets, The Cryosphere, 3, pp. 195-203.

Franke, S., Gerber, T., Warren, C., Jansen, D., Eisen, O., & Dahl-Jensen, D. (2023). Investigating the radar response of englacial debris entrained basal ice units in East Antarctica using electromagnetic forward modelling. IEEE Transactions on Geoscience and Remote Sensing.

Obase, T., Abe-Ouchi, A., Saito, F., Tsutaki, S., Fujita, S., Kawamura, K., and Motoyama, H. (2022): A one-dimensional temperature and age modeling study for selecting the drill site of the oldest ice core around Dome Fuji, Antarctica, The Cryosphere Discussions, 2022, 1–24.

Rodriguez-Morales, F., Braaten, D., Mai, H. T., Paden, J., Gogineni, P., Yan, J.-B., Abe-Ouchi, A., Fujita, S., Kawamura, K., Tsutaki, S., et al. (2020): A Mobile, Multichannel, UWB Radar for Potential Ice Core Drill Site Identification in East Antarctica: Development and First Results, IEEE Journal of Selected Topics in Applied Earth Observations and Remote Sensing, 13, 4836-4847.

| Revision number | Reviewer | Position | Before | Rephrase | Revision | |
|---|---|---|---|---|---|---|
| 1 | 1 2 | Section 2.4 | | | Add further details of the model | |
| 2 | 1 | Line 211 | | | Add the caveats | |
| 3 | 1 2 | Figure 1 | | | Change the contours color of the surface elevation. | |
| 4 | 1 2 | Figure 4 | | | We increase the size of data points. We increase the size of markers of DF and NDF in the figure. We give the values of colorbars at start and end. We put the ticks outside the color bar to see them clearly. | We replace the gray scale bed elevation background with colored contours with different line style (only in Fig. 4a, test). |
| 5 | 1 2 | Figure 5 | | | Add ellipses to show the old ice sites. | Change gray polygon to another color. |
| 6 | 1 2 | Figure 6 | | | Make stagnant ice clearer. | |
| 7 | 1 | Flank flow | | | Rephrase this part and provide velocity map | |
| 8 | 1 2 | Section 2.3 | | | We rephrase this part in a clearer way in revision. | |
| 9 | 1 2 | Section 4.2/4.3.4 | | | We point out the limitation of radar system, i.e., basal unit couldn't be observed in the radargrams. | |
| 10 | 1 | Line 6 | basal layer | stagnant ice | | |
| 11 | 1 | Line 26 | feasible | useful | | |
| 12 | 1 2 | Line 37-39 | In the Dome A region, Sun et al. (2014) estimated ice age around Kunlun station by applying a three dimensional, thermomechanically coupled full-Stokes model, which indicated that in the area without basal melting the ice age at 95 % depth could be limited to 1.5 Ma. | In the Dome A region, Sun et al. (2014) estimated ice age around Kunlun station by applying a three-dimensional, thermomechanically coupled full-Stokes model assuming different geothermal flux and fabrics. They imposed a 1.5 Myr limit to the age solver, thus they didn't get the actual age of the oldest ice, but the distribution of ice potentially older than maximum run time. | | |
| 13 | 1 2 | Line 56- | …Karlsson et al. (2018) presented an updated subglacial topography… | … In the large DF area, Karlsson et al. (2018) presented an updated subglacial topography… | Start a paragraph about the researches in the large Dome Fuji region. Add the conclusions of the previous studies. | |
| 14 | 1 | Line 56 | …based on an airborne radar surveys… | …based on airborne radar surveys… | | |
| 15 | 1 2 | Line 89 | …two-dimension filter… | …running average filter… | | |
| 16 | 1 | Line 98 | We trace 6 or 7 relatively distinct and continuous IRHs (H1--H7) in all survey lines, the third IRH H3 is not clear and continuous enough to be traced in some profiles | We trace 6 (H1, H2, H4 – H7) or 7 (H4 – H7) relatively distinct and continuous IRHs in the radar profiles, since the third IRH H3 is not clear and continuous enough to be traced in some profiles | | |
| 17 | 1 | Line 128 | The estimate of the range precision is always higher than the resolution | The estimate of the range precision is always numerically smaller than the vertical resolution | | |
| 18 | 1 | Line 264 | …We show the reliability index in the… | … We show the reliability index (described in section 2.4) in the… | | |
| 19 | 1 | Line 99 | | Add "This ice thickness data is available on Pangaea (Karlsson et al., 2018)." after "…through semi-automatic detection routines in Matlab…" | | |
| 20 | 1 2 | Figure 1 caption | …from Greene et al. (2017) and Morlighem et al. (2017, 2020) …examplary… | from  Morlighem et al. (2020) and Morlighem (2022) …example… | | |
| 21 | 1 | Section 3.3 | | | Compare the lakes distribution with DC. | |
| 22 | 1 | Figure 7 | | | Use different color to make it clearer. | |
| 23 | 2 | Figure 4d | | | Add figure of age of ice at the depth of 250 m above the bed in results. | |
| 24 | 2 | Section 4.3.3 | | | Rewrite this section. | |
| 25 | 2 | Conclusion | | | Make it more conclusive | |
| 26 | 2 | Figure 3 | | | We give the values of colorbars at start and end. We put the ticks outside the color bar to see them clearly. | |
| 27 | 2 | Grammer | | | Revise carefully. | |
| 28 | 2 | Line 26 | basal layer | deep ice records | | |
| 29 | 2 | Line 70 | a (potentially stagnant) basal layer | the stagnant ice | | |
| 30 | 2 | Line 153 | …there is a basal layer of stagnant ice… | …there is stagnant ice… | | |
| 31 | 2 | Line 367 | basal layer | bottommost part | | |
| 32 | 2 | Figure 4c | | | Add age uncertainty of basal ice as c in the figure. | |
| 33 | 2 | Line 224 | | | Rephrase the paragraph to show the importance of adding the comparison with the previous work. | |
| 34 | 2 | Line 39 | | | Add the reference Beem et al., 2021. | |
| 35 | 2 | Line 82 | The AWI RES system transmits radar waves with a center frequency of 150 MHz and an amplitude of 1.6 kW | The AWI RES system transmits radar waves with a center frequency of 150 MHz, a band width of 20 MHz and an amplitude of 1.6 kW | | |
| 36 | 2 | Figure 2 | | | Remove the black lines. Use plus/minus to replace parenthesis. | |
| 37 | 2 | Line 140 | | | State how accumulation is inferred for ages that predate the oldest ice in the core. | |
| 38 | 2 | Line 141 | inverted | inferred | | |
| 39 | 2 | Line 147 | where $H_m$ is the mechanical ice thickness, which means the effective ice thickness above the stagnant ice, and p is a shape factor controlling vertical deformation (Lilien et al., 2021) | where p is a shape factor controlling vertical deformation (Lilien et al., 2021), $H_m$ is the mechanical ice thickness, which is different to the observed ice thickness $H_{obs}$. When $H_m$ is greater than the observed ice thickness $H_{obs}$, we have melting conditions at the base. Otherwise, there is stagnant ice. If the basal ice is melting, the melt rate m can be obtained by… | | |
| 40 | 2 | Line157/Fig 5 caption/Line 264/Line 265 | reliability index | reliability index $\sigma R$ | | |
| 41 | 2 | Line167/168 | exemplary | example | | |
| 42 | 2 | Line171 | where the mechanical ice thickness Hm (purple dash line) is larger (deeper) than the observed ice thickness (black line) | where the mechanical ice thickness Hm is larger  than the observed ice thickness | | |
| 43 | | Line 6 | …do not exist, the basal thermal conditions, including the thickness of the stagnant ice surface accumulation rates | …do not exist, the surface accumulation rate and the basal thermal condition, including melt rate and the thickness of the stagnant ice. | | |
| 44 | | Line 99 | Matlab | MATLAB | | |
| 45 | | Line 52 | inverted | inferred | | |
| 46 | | Reference | Chung,A.,Parrenin,F.,Steinhage,D.,Mulvaney, R.,Martin,C.,Cavitte,M.,Lilien,D.,Helm,V.,Taylor,D.,Gogineni,P.,Ritz,C.,Frezzotti, M.,O'Neill,C.,Miller,H.,Dahl-Jensen,D.,andEisen,O.:Stagnanticeandagemodel lingintheDomeCregion,Antarctica,submitted. | Chung, A., Parrenin, F., Steinhage, D., Mulvaney, R., Martín, C., Cavitte, M. G. P., Lilien, D. A., Helm, V., Taylor, D., Gogineni, P., Ritz, C., Frezzotti, M., O'Neill, C., Miller, H., Dahl-Jensen, D., and Eisen, O. (2023): Stagnant ice and age modelling in the Dome C region, Antarctica, EGUsphere, 2023, 1–31. | | |
| 47 | | Line 75 | …BaslerBT-67aircraft… | …BaslerBT-67aircraft (Wesche et al., 2016)… | | |
| 48 | | Reference | | Add reference: Wesche, C., Steinhage, D., and Nixdorf, U.: Polar aircraft Polar5 and Polar6 operated by the Alfred Wegener Institute, Journal of large-scale research facilities, 2, 1–7, 2016. | | |
| 49 | | All the orientations | | All directions should be referring to true north, thus, we will add "grid" in front of all the words representing orientations, e.g., we will change Line 180 "to the north-west of the DF drill site" to "to the grid north-west of the DF drill site" | | |

---

## Author Comment (AC3)

Response to RC2 of 'Mapping age and basal conditions of ice in the Dome Fuji region, Antarctica, by combining radar internal layer stratigraphy and flow modeling' (tc-2023-35)

May 19, 2023

We thank Anonymous Referee #2 for forwarding these very helpful and thoughtful review comments. We are most grateful for the time the reviewer spent providing feedbacks on how to improve our manuscript. In our revision, we have tried to address the reviewer's suggestions as much as possible, specified in detail below. For clear tracking of changes, individual issues raised by the referees are referred to as (**Revision I**) below, where "I" is the number of the comment in the attached table.

*This manuscript seeks to map out properties of the ice near the bed around Dome Fuji, East Antarctica. The authors trace layers in an airborne radar survey from 2017/2018, and date those layers using the age scale from an ice core drilled at Dome Fuji proper. They then fit a 1D pseudo-steady model (that has been applied extensively to Dome C) to the isochrones at each trace in the radar survey, from which they get an average accumulation, shape factor, and effective ice thickness (indicating stagnant ice or basal melting) at that point. The results suggest very old ice in the area, albeit with enormous (and, according to the authors, underestimated) uncertainty.*

*In the end, the conclusions here are rather thin. The issue with taking the method of Chung et al., in review, and using it on this survey is that the isochrones used here are much less than half as old (170 ka vs 476 ka), and the resultant impact on the reliability of the results is enormous. Essentially, the issue is that a 170 ka isochrone tells us very little about 1.5 Ma ice. Multiple problems can occur: small violations in model assumptions will result in inferred ages that are unrelated to reality (e.g. if ice flow, ice thickness, or accumulation varied in unexpected ways), and overfitting to measurement errors on these young isochrones will cause incorrect results at older ages. Indeed, when the authors check this possibility with even a 230 ka isochrone, they find that there is a huge change in ages as we would expect if the young isochrones simply do not carry much information about deeper ages. In my view, that this problem is occurring is demonstrated conclusively since the model does not match the Dome Fuji ice core's age scale. This is the only really available test of the model reliability—and not only does the model miss the age scale, but it does so outside its reported uncertainty! That is to say, the model is both wrong and confidently so. As a result, I do not think that much can be concluded from this paper, other than that old ice may or may not exist at Dome F. To their credit, the authors describe the limitations in the discussion and conclusion (though it should be better disclosed in the results), so I think that after revision the work will be publishable in The Crysosphere.*

*I think the presentation could be improved in both the paragraph-scale structure and at the sentence level, and the figures could be improved as well. The paper is in need of quite a bit of grammatical work, as there are a lot of missing or extra articles and some subject/verb mismatch. I did not enumerate these in my review. At times I found the paper hard to read, although I think I could eventually discern the meaning, so this could probably be handled by a copy editor. At some points, paragraphs wander away from the thread of the manuscript (see general comments). I agree with the other reviewer that the figures are pretty difficult to make out given their size. Throughout the manuscript, colorbars start and end at arbitrary values and demarcations are difficult to read—well chosen start and end values, arrows to indicate whether the colorbar values are inclusive, and enlargement would help.*

To avoid the misunderstanding of the dataset we have used in this study, it's important to clarify the radar dataset was collected with the AWI radio-echo sounding (RES) system during 2016-2017 Antarctica season, not the dataset collected with the UWB radar in 2017-2018 season. As stated in the system description, this is a pulse system which records rectified waveforms, i.e., it does not record any phase information, thus yielding lower resolution, especially of internal layers.

Because of the data quality, the continuous internal reflection horizons (IRHs) can only be traced in the upper part and dated back to 170 ka. This limitation couldn't be solved from the data side. We agree with the comment that the unexpected variations of thinning in the deep part could lead to the mismatch between the model and reality. As the reviewer pointed out, there is a significant overestimation of age of basal ice at DF (Figure 7), however, we think that this figure actually proved the reliability of the model to a certain extent.

In Figure 7, we show the age-depth scale derived from models at Dome Fuji. We pointed out the age of basal ice was overestimated by a factor of 2 at DF. This figure shows, for RUN II (the run with 7 IRHs), that the deepest IRH was traces at the depth of 2238.22 m at DF, and dated back to 232.65 ka BP. We consider that the model has a reasonable performance (agreement with the timescale DFO2006+AICC2012) to the depth of 2759 m, which is dated back to 540.48 ka BP. This depth is about 290 m above the bed, which is exactly the depth of the inflection point in the timescale DFO2006+AICC2012. As a model assumption, the age-depth profile of the model follows the exponential distribution below this depth. Thus, the large overestimation is actually caused by the curvature reversal below this depth in the timescale of the ice core.

A similar phenomenon was also observed in our companion study (Chung et al. in review) at EDC though much older IRHs were dated there. They pointed out: "The modelled age at the deepest dated point for the EDC drill site which was around 100-200 kyr older than would be expected from the AICC2012 age-depth profile ... Looking at the AICC2012 profile determined by experimental measurements, it follows an exponential profile until the lower 200 m of dated ice, perhaps meaning that the thinning is for some reason lower than the model would expect…".

Figure R1 here simply shows the comparison between the age-depth profile derived by the model and the timescale at EDC (AICC2012). The model result agree with the timescale perfectly until the timescale deviates from the exponential form. At DC the overestimation is more reasonable since the timescale at DC doesn't change as drastically as that at DF. In the DF case, to solve this overestimation problem, only more continuous isochrones below the inflection point, i.e., the lowest 300 m could provide better constraint for the model, which is not possible in our dataset, and in fact most likely also not easily possible with other data sets (Tsutaki et al., 2022).

The drilling of the DF deep ice core indicated melting at DF (Motoyama et al., 2021), which could explain the inflection point in the age-depth scale of the ice core. This would imply that the significant overestimation likely occurs in the area with basal melting. Given that there is only one deep ice core in the DF region, we lack an additional timescale extending towards the bedrock to prove our hypothesis. At the same time, we find it unjustified to draw the conclusion that the model approach does not work in the DF region. According to what we have found by now, the model works quite well in the upper 2/3 of the ice column by calculating our reliability index (the standard deviation of the age difference between

observation and model results), and it also seems appropriate in the deeper part at DF, until reaching the depth of the inflection point of timescale.

Therefore, we proposed "the reliability of our model is probably overestimated" in the manuscript as a responsible statement. In fact, this probability is higher in the area with basal melting. We will highlight this hypothesis and emphasize the importance of considering basal thermal state while finding old ice by our model more in the revision. Since we have found that melting prevails over stagnant ice in the DF region by using our model approach, it is possible that significant overestimations of age occur in the deep ice in the ice) at various places, as what has been observed at the DF. Taking this into consideration, we will include an additional Figure 4d in the manuscript that depicts the age of ice at a depth of 250 m above the bed. This figure could provide relatively accurate age values while excluding the lowest part of the ice. **(Revision 23)**

[Figure]

**Figure R1. Model derived age-depth scale and AICC2012 timescale at EDC.**

In the revised version of the manuscript, we will improve the section 4.3.3 in the sense of what we have discussed here. **(Revision 24)**

As the reviewer suggested, we will also improve our conclusion. **(Revision 25)**

We will follow the comments and refine our figures by adding the start and end values to the color bars and placing the ticks outside the color bars. **(Revision 3-6, 26)**

We will conduct a thorough review of the manuscript, carefully addressing and correcting the grammar and vocabulary errors. **(Revision 27)**

*General comments*
*Basal layer is a sticky term. As used on line 26, it sounds like generic deep ice. However, in other literature, it refers specifically to ice that has a distinct radar character, to the ice near the bottom of EDC that has little discernible paleoclimatic information, or to ice that is inferred to be stagnant. The situation is further muddled on line 70, where the authors suggest that their method can detect a potentially stagnant basal layer—but this is incorrect. The method can only detect a stagnant basal layer (at least in terms of vertical velocity), since the whole premise of the detection is that the basal layer is stationary for the purposes of the depth-age scale. This work should be consistent on its usage of the term basal layer and it should define what it means by that in the introduction. Imprecise usages such as that in line 26 should be removed. Some discussion should be added on whether there is any correspondence between the areas where stagnant ice is inferred and any characteristic of the radargrams.*

Thanks for pointing out the confusing use of 'basal layer'. We decide to use 'basal unit' as the bottommost part (which has been used in the literature before) in which there are some peaks in return power but no continuous or coherent reflecting horizons. We use "stagnant ice" as the stagnant bottommost ice above the ice-bed interface derived from the model.

We will change "basal layer" to "deep ice records" in Line 26. **(Revision 28)**

We will change "a (potentially stagnant) basal layer" to "the stagnant ice" in Line 70. **(Revision 29)**

Except for the two changes suggester by reviewer, we will also change "...there is a basal layer of stagnant ice…" to "...there is stagnant ice…" in Line 153. **(Revision 30)**

We will change "basal layer" to "bottommost part" in Line 367. **(Revision 31)**

In the radar dataset we used, there is an echo free zone (EFZ) above the bed, with a thickness of several hundreds of meters. EFZ could be caused by various reasons, e.g., system sensitivity of the radar system, deformation, recirculation and recrystallization of ice (Drews et al., 2009 and Franke et al., 2023). We think the EFZ in our dataset could most likely be caused by the performance of the radar system, since in the same region more modern radar can detect somewhat deeper, more coherent horizons, as we have pointed out in our manuscript (L340), 'It implied that at the same depth modern systems would provide not only a higher resolution, but most likely also a deeper detection of continuous IRHs (Rodriguez-Morales et al., 2020).' We will point this out in section 4.2 when we discuss the basal thermal state and 4.3.4 when we discuss the limitation of radar system. **(Revision 9)**

*Given the wide pulses of this radar system, and the lack of pulse decompression, I am skeptical of the vertical precision that the authors are claiming and I would suspect a bias. Just by two way travel time, the pulse is 50 m long (as the authors correctly point out, this is the resolution, defined here as the separation needed to identify two targets as distinct). Again, as the authors correctly identify, this resolution is different than the precision (i.e. the depth-accuracy of a target). However, the authors trace in a fairly standard manner (picking*

*the strongest return), but given the processing of these data the depth of the reflector should really be off the first return, assuming that time zero is defined as the start of the pulse (indeed, this is what you would get if you could do pulse decompression). The problem is that I would expect the strongest return to lie below the first return from a reflector (most likely 25 m below, but this offset is somewhat arbitrary), but never above. Thus, I think that there is likely a systematic bias in the ages and depths used. This may have a small effect on age-depth scales in the end, since it may affect an isochrone in the same manner along its length, but it should be accounted for carefully.*

In the next figures R2 and R3 we show some screenshots of the returns of different IRHs, the horizontal axis shows traces, the vertical axis shows Time ($\mu s$) (the unit is $s$ on the figure, because the software is originally used for seismic data).

[Figure]

**Figure R2. Zoom-in view of IRH4 returns in Profile 20172029.**

For a single pulse, the bias between the first return and the strongest return varies, but generally ranges from 50-200 ns.

[Figure]

**Figure R3. Zoom-in view of IRH7 returns in Profile 20172029.**

For IRH7, in most traces, multiple return pulses overlap. E.g., for trace 22815, the first return occurs at 33430 ns, but the strongest return we traced in Paradigm could be at 33850 ns, what

makes the offset around 420 ns. The bias varies from 50-420 ns in the best case, which is too big to be a systematic bias regarding the pulse length of 600 ns.

Hence, we prefer to keep our revised description for the age uncertainty. **(Revision 8)**

*Overall, uncertainty deserves a more prominent place in the results and discussion. First, how is the basal age uncertainty that the authors report calculated? I am guessing it is as in Chung et al., but there is not even a passing mention in this manuscript of how the authors obtain anything other than a best-fit value—it is critical to add this to the manuscript. Then, there is the issue of uncertainty in basal age ice—the number reported for Dome F itself is plus minus 500 kyr. This is enormous, over a third of the age. Figure 4 should plot the uncertainty on the basal ages—without this, the reader has no idea if there are any areas at all with reliably old ages. This gets addressed later by table 1 and the sensitivity analysis, but I see no reason that it cannot fit in Figure 4.*

Thanks for pointing this out. The theory is the same as the one described in Chung et al., which provides a detailed account of the approach. Nevertheless, we will add more details of the age uncertainty at the end of the Method session to make it clearer. **(Revision 1)**

We consider the value of the uncertainty of the basal age as a significant quantity in the entire region. We have compared it with that from the Dome C region. We gave the age uncertainty almost the same order of magnitude for each isochrone, but the basal age uncertainty is much smaller in the Dome C region. We relate this to the fact that the number of isochrones used as constraints for the model is larger in the Dome C region. We will add the age uncertainty of the basal ice as Figure 4c. **(Revision 32)**

*The comparison with Karlsson 2018 and van Liefferinge 2018 deserves more consideration. While the approaches are different, what can we learn by comparing them? Where should we believe their results over the results here (for example, are the results here thermodynamically tenable)? Where is there agreement?*

Karlsson et al. (2018) and van Liefferringe et al., (2018) used a thermodynamical model to identify old ice sites, but their approaches did not include any constraint from the age of radar internal layers. The model we are using does not take into account the thermal dynamics at all, thus being more independent of GHF estimates. However, the sites with potentially "old ice" suggested by the different approaches show a considerable correspondence in some places, especially around DF and NDF. We think that the underlaying reason for this consistency is the use of ice thickness in both models, and the crucial impact of ice thickness on the age distribution of the ice.

In the figure, we put the sites with old ice suggested by the thermal model on top of our results. In this way we aim at giving readers a visual impression, which site is more likely to hold old ice, both mechanically and thermodynamically. We will rephrase the paragraph to make it more logical. **(Revision 33)**

*The introduction could use some work. We would benefit from more focus on what the reader should take away. This is most obvious in the paragraph beginning at line 40 wanders between detailed analysis of timescales, studies that concluded something about basal thermal state and potential old ice sites (without ever stating those conclusions). I suggest a careful culling of the introduction, focusing on the goals of each paragraph.*

We will split the paragraph to two parts, the research conducted on the Dome Fuji ice core and other studies performed in the large Dome Fuji region. We will add the conclusions of the previous studies. **(Revision 13)**

*L36: If efforts beyond BE-OI are discussed, it seems strange to include Dome A but not other countries' oldest ice efforts. For example, Beem et al. (2021) would then also be appropriate here. Also on L36, the sentence needs to be rephrased—is the point that the age is limited or that it reaches 1.5 Ma?*

We will add the reference as suggested to have a complete overview. **(Revision 34)**

We will change Line 36-39 "In the Dome A region, Sun et al. (2014) estimated ice age around Kunlun station by applying a three-dimensional, thermomechanically coupled full-Stokes model, which indicated that in the area without basal melting the ice age at 95 % depth could be limited to 1.5 Ma."
to
"In the Dome A region, Sun et al. (2014) estimated the age of ice around Kunlun station by applying a three-dimensional, thermomechanically coupled full-Stokes model assuming different geothermal flux and fabrics. They imposed a 1.5 Myr limit to the age solver, thus they did not get the actual age of the oldest ice, but the distribution of ice potentially older than maximum run time of their model." **(Revision 12)**

*L81: I find this system description to be a bit vague. I was under the impression that the AWI system is multi-channel and phase-coherent. Is this a single channel power, or the total transmit power? Pulse-limited is not a term we see that often for ice-penetrating radars (perhaps more for radar altimeters), so it would be helpful to say the importance explicitly— that it acts much like a chirp system where you cannot decompress the chirps (thus the very low vertical resolution). Is there no reported bandwidth because there was no frequency sweep in the chirp? If the chirp did sweep frequencies, the bandwidth should be reported.*

The dataset we have used in this research was collected by AWI radio-echo sounding (RES) system during 2016-2017 Antarctica season, not the dataset collected by UWB radar in 2017-2018 season.

The AWI RES system is a burst system, with center frequency of 150 MHz. The pulse generated has length of 60 ns or 600 ns, as we referenced in our manuscript (Nixdorf et al., 1999). The signal transmitted by the system is not chirp with a frequency sweep, but a short, high-power pulses with specific widths, which is in addition rectified after reception. We will also mention the bandwidth in Line 82, we will change
"The AWI RES system transmits radar waves with a center frequency of 150 MHz and an amplitude of 1.6 kW."
to
"The AWI RES system transmits radar waves with a center frequency of 150 MHz, a band width of 20 MHz and an amplitude of 1.6 kW." **(Revision 35)**

*L89: Should explicitly state what kind of 2d filter*

It is a particular filter in the software "Echos", which we use for analysis. Since the filter could be adapted to the radar data in both trace direction and time direction, it is called a 2D filter. This filter returns a weighted running average. It is summing up amplitudes at the same travel-time, which could remove the horizontal noise. We will change

"…a low-pass filter and a two-dimension filter are…"

to

"…a low-pass filter and a running average filter are…" **(Revision 15)**.

*Figure 1: "examplary" implies that it is an ideal or "the best" profile, while I think simply "example" would be more accurate. Different colors for contours and survey would help readabilility. An increase in size would be nice too.*

We will change "exemplary" to "example". **(Revision 20)**

We will change the color of the contours. **(Revision 3)**

*Figure 2: The horizontal black lines make this almost impossible to interpret. I would also like to see a zoom in on some of the tracing, so that we can see how the traced depth relates to the width of the returns. Why not just use a plus/minus for the uncertainty and avoid the confusing double parenthetical?*

We will remove the black lines and improve the figure.

[Figure]

**Figure R4. Zoom-in view of IRH4 picked in Profile 20172040.**

We show a zoom-in view of profile 20172040 in Fig. R4. Same as the returns of zoom-in view of different horizons in Profile 20172029, which we have shown in General comment, there is no trustworthy systematic bias that could be adapted to all the picks.

Thanks for the comment, we will use plus/minus to replace double parenthetical. **(Revision 36)**

*L140: This needs to state how accumulation is inferred for ages that predate the oldest ice in the core*

Thanks, we will state how accumulation rate inferred. **(Revision 37)**

*L141: "Inverted" means "was turned upside down"—but this was both inverted and integrated. Here "inferred" or simply "calculated" would be correct.*

Thanks, we will change "inverted" to "inferred". **(Revision 38)**

*L142: This needs to be rephrased to make clear that the presence of stagnant ice is undetermined*

Thanks for the comment. From Line 147, we reorganize the paragraphs, by moving the description of $H_m$ forward, and then describe the optimization.

We will change
"…where $H_m$ is the mechanical ice thickness, which means the effective ice thickness above the stagnant ice, and $p$ is a shape factor controlling vertical deformation (Lilien et al., 2021)."
to
"…where $p$ is a shape factor controlling vertical deformation (Lilien et al., 2021), $H_m$ is the mechanical ice thickness, which is different to the observed ice thickness $H_{obs}$. When $H_m$ is greater than the observed ice thickness $H_{obs}$, we have melting conditions at the base. Otherwise, there is stagnant ice. If the basal ice is melting, the melt rate $m$ can be obtained by…" **(Revision 39)**

*L149: Considering that Chung et al. And Lilien et al. use inverse methods that produce uncertainties, it is important to state here whether this method does the same or whether this work simply finds the single solution with the lowest misfit.*

We will add a few paragraphs to describe the optimization methods. **(Revision 1)**

*L156: Could this be renamed? It is essentially the inverse of the reliability, so it is quite confusing to call it the reliability index.*

For consistence with our companion manuscript, we prefer to keep the same naming. But to avoid misunderstanding, we add "$\sigma_R$" after "reliability index". **(Revision 40)**

*3.1: Same issue with meaning of "exemplary"*

We will change "exemplary" to "example" in Line 167 and 168. **(Revision 41)**

*L171: Delete parenthetical—thickness cannot be deep*

Done. **(Revision 42)**

*Figure 5: The Van Liefferinge data need a different color—gray on a gray background is unreadable.*

We will change the gray polygons to (another) color. **(Revision 5)**

*Final paragraph: I do not think this is a conclusion, nor is it a logical way to approach the limitations of this work. Overall, it is a rather weak note to end on—why not move it to the discussion, which is what it really is anyway?*

We will rewrite the conclusion. **(Revision 25)**

**References**

Chung, A., Parrenin, F., Steinhage, D., Mulvaney, R., Martín, C., Cavitte, M. G. P., Lilien, D. A., Helm, V., Taylor, D., Gogineni, P., Ritz, C., Frezzotti, M., O'Neill, C., Miller, H., Dahl-Jensen, D., and Eisen, O. (2023): Stagnant ice and age modelling in the Dome C region, Antarctica, EGUsphere, 2023, 1–31.

Drews, R., Eisen, O., Weikusat, I., Kipfstuhl, S., Lambrecht, A., Steinhage, D., Wilhelms, F. and Miller, H. (2009): Layer disturbances and the radio-echo free zone in ice sheets, The Cryosphere, 3, pp. 195-203.

Franke, S., Gerber, T., Warren, C., Jansen, D., Eisen, O., & Dahl-Jensen, D. (2023). Investigating the radar response of englacial debris entrained basal ice units in East Antarctica using electromagnetic forward modelling. IEEE Transactions on Geoscience and Remote Sensing.

Karlsson, N. B., Binder, T., Eagles, G., Helm, V., Pattyn, F., Van Liefferinge, B., and Eisen, O. (2018): Glaciological characteristics in the Dome Fuji region and new assessment for "Oldest Ice", The Cryosphere, 12, 2413–2424.

Lilien, D. A., Steinhage, D., Taylor, D., Parrenin, F., Ritz, C., Mulvaney, R., Martín, C., Yan, J.-B., O'Neill, C., Frezzotti, M., Miller, H., Gogineni, P., Dahl-Jensen, D., and Eisen, O. (2021): Brief communication: New radar constraints support presence of ice older than 1.5 Myr at Little Dome C, The Cryosphere, 15, 1881–1888.

Motoyama, H., Takahashi, A., Tanaka, Y., Shinbori, K., Miyahara, M., Yoshimoto, T., Fujii, Y., Furusaki, A., Azuma, N., Ozawa, Y., et al. (2021): Deep ice core drilling to a depth of 3035.22 m at Dome Fuji, Antarctica in 2001–07, Annals of Glaciology, 62, 212–222.

Nixdorf, U., Steinhage, D., Meyer, U., Hempel, L., Jenett, M., Wachs, P., and Miller, H. (1999): The newly developed airborne radio-echo sounding system of the AWI as a glaciological tool, Annals of Glaciology, 29, 231–238.

Rodriguez-Morales, F., Braaten, D., Mai, H. T., Paden, J., Gogineni, P., Yan, J.-B., Abe-Ouchi, A., Fujita, S., Kawamura, K., Tsutaki, S., et al. (2020): A Mobile, Multichannel, UWB Radar for Potential Ice Core Drill Site Identification in East Antarctica: Development and First Results, IEEE Journal of Selected Topics in Applied Earth Observations and Remote Sensing, 13, 4836-4847.

Tsutaki, S., Fujita, S., Kawamura, K., Abe-Ouchi, A., Fukui, K., Motoyama, H., Hoshina, Y., Nakazawa, F., Obase, T., Ohno, H., Oyabu, I., Saito, F., Sugiura, K., and Suzuki, T. (2022): High-resolution subglacial topography around Dome Fuji, Antarctica, based on ground-based radar surveys over 30 years, The Cryosphere, 16, 2967–2983.

Van Liefferinge, B., Pattyn, F., Cavitte, M. G., Karlsson, N. B., Young, D. A., Sutter, J., and Eisen, O. (2018): Promising Oldest Ice sites in East Antarctica based on thermodynamical modelling, The Cryosphere, 12, 2773–2787.

| Revision number | Reviewer | Position | Before | Rephrase | Revision | |
|---|---|---|---|---|---|---|
| 1 | 1 2 | Section 2.4 | | | Add further details of the model | |
| 2 | 1 | Line 211 | | | Add the caveats | |
| 3 | 1 2 | Figure 1 | | | Change the contours color of the surface elevation. | |
| 4 | 1 2 | Figure 4 | | We increase the size of data points. We increase the size of markers of DF and NDF in the figure. We give the values of colorbars at start and end. We put the ticks outside the color bar to see them clearly. | We replace the gray scale bed elevation background with colored contours with different line style (only in Fig. 4a, test). | |
| 5 | 1 2 | Figure 5 | | | Add ellipses to show the old ice sites. | Change gray polygon to another color. |
| 6 | 1 2 | Figure 6 | | | Make stagnant ice clearer. | |
| 7 | 1 | Flank flow | | | Rephrase this part and provide velocity map | |
| 8 | 1 2 | Section 2.3 | | | We rephrase this part in a clearer way in revision. | |
| 9 | 1 2 | Section 4.2/4.3.4 | | | We point out the limitation of radar system, i.e., basal unit couldn't be observed in the radargrams. | |
| 10 | 1 | Line 6 | basal layer | stagnant ice | | |
| 11 | 1 | Line 26 | feasible | useful | | |
| 12 | 1 2 | Line 37-39 | In the Dome A region, Sun et al. (2014) estimated ice age around Kunlun station by applying a three dimensional, thermomechanically coupled full-Stokes model, which indicated that in the area without basal melting the ice age at 95 % depth could be limited to 1.5 Ma. | In the Dome A region, Sun et al. (2014) estimated ice age around Kunlun station by applying a three-dimensional, thermomechanically coupled full-Stokes model assuming different geothermal flux and fabrics. They imposed a 1.5 Myr limit to the age solver, thus they didn't get the actual age of the oldest ice, but the distribution of ice potentially older than maximum run time. | | |
| 13 | 1 2 | Line 56- | …Karlsson et al. (2018) presented an updated subglacial topography… | … In the large DF area, Karlsson et al. (2018) presented an updated subglacial topography… | Start a paragraph about the researches in the large Dome Fuji region. Add the conclusions of the previous studies. | |
| 14 | 1 | Line 56 | …based on an airborne radar surveys… | …based on airborne radar surveys… | | |
| 15 | 1 2 | Line 89 | …two-dimension filter… | …running average filter… | | |
| 16 | 1 | Line 98 | We trace 6 or 7 relatively distinct and continuous IRHs (H1--H7) in all survey lines, the third IRH H3 is not clear and continuous enough to be traced in some profiles | We trace 6 (H1, H2, H4 – H7) or 7 (H4 – H7) relatively distinct and continuous IRHs in the radar profiles, since the third IRH H3 is not clear and continuous enough to be traced in some profiles | | |
| 17 | 1 | Line 128 | The estimate of the range precision is always higher than the resolution | The estimate of the range precision is always numerically smaller than the vertical resolution | | |
| 18 | 1 | Line 264 | …We show the reliability index in the… | … We show the reliability index (described in section 2.4) in the… | | |
| 19 | 1 | Line 99 | | Add "This ice thickness data is available on Pangaea (Karlsson et al., 2018)." after "…through semi-automatic detection routines in Matlab…" | | |
| 20 | 1 2 | Figure 1 caption | …from Greene et al. (2017) and Morlighem et al. (2017, 2020) …examplary… | from  Morlighem et al. (2020) and Morlighem (2022) …example… | | |
| 21 | 1 | Section 3.3 | | | Compare the lakes distribution with DC. | |
| 22 | 1 | Figure 7 | | | Use different color to make it clearer. | |
| 23 | 2 | Figure 4d | | | Add figure of age of ice at the depth of 250 m above the bed in results. | |
| 24 | 2 | Section 4.3.3 | | | Rewrite this section. | |
| 25 | 2 | Conclusion | | | Make it more conclusive | |
| 26 | 2 | Figure 3 | | | We give the values of colorbars at start and end. We put the ticks outside the color bar to see them clearly. | |
| 27 | 2 | Grammer | | | Revise carefully. | |
| 28 | 2 | Line 26 | basal layer | deep ice records | | |
| 29 | 2 | Line 70 | a (potentially stagnant) basal layer | the stagnant ice | | |
| 30 | 2 | Line 153 | …there is a basal layer of stagnant ice… | …there is stagnant ice… | | |
| 31 | 2 | Line 367 | basal layer | bottommost part | | |
| 32 | 2 | Figure 4c | | | Add age uncertainty of basal ice as c in the figure. | |
| 33 | 2 | Line 224 | | | Rephrase the paragraph to show the importance of adding the comparison with the previous work. | |
| 34 | 2 | Line 39 | | | Add the reference Beem et al., 2021. | |
| 35 | 2 | Line 82 | The AWI RES system transmits radar waves with a center frequency of 150 MHz and an amplitude of 1.6 kW | The AWI RES system transmits radar waves with a center frequency of 150 MHz, a band width of 20 MHz and an amplitude of 1.6 kW | | |
| 36 | 2 | Figure 2 | | | Remove the black lines. Use plus/minus to replace parenthesis. | |
| 37 | 2 | Line 140 | | | State how accumulation is inferred for ages that predate the oldest ice in the core. | |
| 38 | 2 | Line 141 | inverted | inferred | | |
| 39 | 2 | Line 147 | where $H_m$ is the mechanical ice thickness, which means the effective ice thickness above the stagnant ice, and p is a shape factor controlling vertical deformation (Lilien et al., 2021) | where p is a shape factor controlling vertical deformation (Lilien et al., 2021), $H_m$ is the mechanical ice thickness, which is different to the observed ice thickness $H_{obs}$. When $H_m$ is greater than the observed ice thickness $H_{obs}$, we have melting conditions at the base. Otherwise, there is stagnant ice. If the basal ice is melting, the melt rate m can be obtained by… | | |
| 40 | 2 | Line157/Fig 5 caption/Line 264/Line 265 | reliability index | reliability index $\sigma R$ | | |
| 41 | 2 | Line167/168 | exemplary | example | | |
| 42 | 2 | Line171 | where the mechanical ice thickness Hm (purple dash line) is larger (deeper) than the observed ice thickness (black line) | where the mechanical ice thickness Hm is larger  than the observed ice thickness | | |
| 43 | | Line 6 | …do not exist, the basal thermal conditions, including the thickness of the stagnant ice surface accumulation rates | …do not exist, the surface accumulation rate and the basal thermal condition, including melt rate and the thickness of the stagnant ice. | | |
| 44 | | Line 99 | Matlab | MATLAB | | |
| 45 | | Line 52 | inverted | inferred | | |
| 46 | | Reference | Chung,A.,Parrenin,F.,Steinhage,D.,Mulvaney, R.,Martin,C.,Cavitte,M.,Lilien,D.,Helm,V.,Taylor,D.,Gogineni,P.,Ritz,C.,Frezzotti, M.,O'Neill,C.,Miller,H.,Dahl-Jensen,D.,andEisen,O.:StagnanticeandagemodellingintheDomeCregion,Antarctica,submitted. | Chung, A., Parrenin, F., Steinhage, D., Mulvaney, R., Martín, C., Cavitte, M. G. P., Lilien, D. A., Helm, V., Taylor, D., Gogineni, P., Ritz, C., Frezzotti, M., O'Neill, C., Miller, H., Dahl-Jensen, D., and Eisen, O. (2023): Stagnant ice and age modelling in the Dome C region, Antarctica, EGUsphere, 2023, 1–31. | | |
| 47 | | Line 75 | …BaslerBT-67aircraft… | …BaslerBT-67aircraft (Wesche et al., 2016)… | | |
| 48 | | Reference | | Add reference: Wesche, C., Steinhage, D., and Nixdorf, U.: Polar aircraft Polar5 and Polar6 operated by the Alfred Wegener Institute, Journal of large-scale research facilities, 2, 1–7, 2016. | | |
| 49 | | All the orientations | | All directions should be referring to true north, thus, we will add "grid" in front of all the words representing orientations, e.g., we will change Line 180 "to the north-west of the DF drill site" to "to the grid north-west of the DF drill site" | | |

---

## Author Response (AR1)

**Response Letter to Reviewers**

**Response to RC 1 of 'Mapping age and basal conditions of ice in the Dome Fuji region, Antarctica, by combining radar internal layer stratigraphy and flow modeling' (tc-2023-35)**

*The paper provides an assessment of the distribution of ice that meets the targets of the International Partnerships in Ice Core Science "Oldest Ice Challenge" using a kinematic approach inverting observed englacial isochrons dated using an ice core pinning point for the deep age structure of the ice sheet. This 1-D approach (specifically IsoInv) has been primarily used for the Dome C region, and stands in contrast to 1D thermodynamic approaches that balance both mass and heat through latent heat (eg van Liefferinge et al., 2014, 2018), and 3D full stokes modeling. The approaches are complementary, with their strengths and weaknesses - thermodynamic approaches have to deal with the tremendous uncertainty in the value of geothermal heat flux, while attaining the deep dated isochrons needed for IsoInv is fraught and the computational cost and rheological and boundary condition uncertainties for 3D approaches limit their usefulness.*
*This paper represents an overall readable contribution to the literature on old ice distribution modeling. I don't have fundamental problems with the conclusion. Aspects of the presentation and discussion would benefit from being made clearer.*

We thank Anonymous Referee #1 for taking the time and effort to provide us with the helpful comments. We believe the manuscript will be improved by following these comments. For clear tracking of changes, individual issues raised by the referees are referred to as (**Revision I**) below, where "I" is the number of the comment in the attached table (to provide an overview). The line number of each revision could be found in the table.

*Major points:*
*Gray literature: The paper does lean on preprints somewhat. The study by Chung et al., submitted is obviously highly coupled with many of the same authors - one would anticipate that the methods this paper refers to will not change greatly, but there is a risk. A larger issue is the pointers to the preprint by Obase et al. 2022, looking at complementary thermodynamic modeling where the reviews indicate that there was an error in the calculation of the basal thermal gradient. Presumably this is addressed in the final accepted (but not yet published) version of that paper. While the Obase paper is obviously very complementary, the Wang paper should either drop the explicit comparisons, or at least provide some caveats on that comparison.*

We thank the reviewer for pointing out the gray literatures.

- The study by Chung et al. (2023, which is now in press) is our companion manuscript, the IsoInv model was used in both studies, and Chung et al. gave a detailed description of the model in their manuscript. In our manuscript, we cited their research for either directing people who are more interested in the model towards their study or comparing the basal thermal state and model reliability in the DF and DC region. Nevertheless, we have added further details of the model, i.e., the optimization algorithm in the Method section. Therefore, we made this part more independent from our companion research. **(Revision 1)**
- The final revised Obase et al. (2022) paper is available online now. We have rephrased the comparison in our manuscript regarding their revisions of model results **(Revision 2)**.

*Figures:* *The maps are hard to read, especially at printed scale. I recommend removing the greyscale DEM bed elevation background and using some well spaced contours instead in a different color than the surface contours. Increasing the plotted point size of the data point may also help make the ideas conveyed more visible. The authors discuss specific candidate old ice sites in the text - they should indicate them on at least one of the maps. The indicators for DF and NDF are hard to make out at printed scale. More specific points are below.*

Thanks for the suggestions on figures. We follow the suggestions and replot the Fig 1, 4, 5, 6.
**(Revision 3, 4, 5, 6)**

- We replaced the gray scale bed elevation background with colored contours with different line style in Figure 4a. We have not adapted this change to other figures yet since we found the corrected one seems less readable. We put the new Figure 4 here as the example, the left one uses the contour to replace the gray background. It is very difficult to be concise and at the same time be very inclusive and take into account the needs of colour blind people. We tried to to that but at a certain point a limit is reached. Here we would kindly ask the editor and reviewer for guidance whether this is sufficient or which way to follow further.

[Figure]

**Figure R1.** (a) Modelled age of the basal ice at a maximum age density of 20 ka m$^{-1}$. (b) Depth of the basal ice at an age density of 20 ka m$^{-1}$.

- We have doubled the size of data points.
- We have added four ellipses to Fig. 5a (Age density of ice at 1.5 Ma) to clarify the locations of old ice candidates. **(Revision 5)**
- We have doubled the size of markers of DF and NDF in the figures.

*Flank flow versus divide flow:* *1-D models are really only appropriate where ice velocities are very low, as the authors point out, but the case they make for Dome F could be made better. They use the statistical spread of basal ice ages as a function of distance from key Dome points as an indicator for flank flow, but don't make their logic clear on why that should be the case. Isn't this statistical trend just a fractal distribution as you cover more and more area with expanding range from the dome? A map of ice flow velocity, or an indication of ice velocity on the maps would be helpful.*

We agree that 1D is appropriate where ice velocities are low. But how low? That's what we would like to illustrate with this analysis. As the distributions are normalized with respect to depth, they should in principle be independent of ice depth and flank flow and thus overlap more strongly than if those regimes are included, where flank flow becomes more dominant. As we consider this a useful and simple illustration of the underlying logic, which we have made clearer in the revision, we can also consider to discard this analysis if the editor or reviewer consider it too far from the main case. We provided a map with velocities in Fig. 1b in the revision. **(Revision 7)**

*Age uncertainties of internal reflection horizons:* *This was confusing. How are depth uncertainty, range precision and best guess uncertainty combined? Is range precision actually calculated from using the SNR? If so, where are the results (which should be different for each IRH).*

Sorry for the confusing description. People have used different methods to evaluate the uncertainty of horizon positions. Lilien et al. (2021) used quarter-wavelength uncertainty and Cavitte et al. (2016) use a concept of range estimate precision. In our research, we follow the latter method, because we want to involve the impact from sub-resolution of different horizons.

The range estimate precision in determining reflection depth is determined by the pulse width of the radar waveform, the signal-to-noise ratio and the sub-resolution reflector fluctuations. The last term could be ignored when the reflections have "continuality of reflection amplitudes and subsequent traceability" (Cavitte et al., 2016), but this is not the case in our study.

We use radar data collected by AWI RES system with 600 ns in this study, which means ~25 m vertical resolution in the radargram. The resolution is lower than that of more advanced radar systems, which causes lower subsequent traceability. Moreover, the bedrock topography is characterized by a series of mountain ranges and valleys and wide melting distribution in the Dome Fuji region, which leads to the discontinuity of isochrones at some places, especially near the bottom. These reasons mentioned above could explain that we need to consider the sub-resolution of different reflectors.

We found that the uncertainty caused by the low traceability and continuity is actually large when we traced the horizons semi-automatically, which means we need to choose where to trace the horizons in the disturbed discontinuous places with a few possible trace routes. E.g., in the radar segment shown below, the horizon in the blue frames is easy to be traced, but between the frames, the stratigraphy is harder to follow and we need to interpolate it referring to the shape and pattern of other internal layers. This is a standard approach for instance also used in marine geophysics to interpolate stratigraphic boundaries. The uncertainty of

different isochrones ranges from 20 m to 50 m for different horizons. We call it best-guess uncertainty. In our case, the range estimate precision calculated from the SNR and range resolution is almost the same with the range resolution (~25 m), but the precision is actually lower in the deep ice, so we finally took the best-guess uncertainty as the uncertainty of horizon positions. To conclude, the tracing process brought larger uncertainty with it, mainly because of the resolution of radar system, compared to radar systems used in other research. This large uncertainty needs to be estimated for different horizons during tracing, and considered into the uncertainty part.

We have rephrased this part in a clearer way in the revision. **(Revision 8)**

[Figure]

**Figure R2.** A segment of a radargram showing the discontinuous isochrone.

*Stagnant Ice:  At Dome C the stagnant layer has a distinctive radar character.  The authors should comment either if similar features are seen at Dome Fuji, or if the radars that have been used can even detect it.*

In the radar dataset we used, there is an echo free zone (EFZ) above the bed, with a thickness of several hundreds of meters. EFZ could be caused by a several reasons, e.g., system sensitivity of the radar system, deformation and folding, or recrystallization of ice (Drews et al., 2009 and Franke et al., 2023). We consider that the EFZ in our dataset could be caused by the performance of the radar system, since in the same region more sophisticated radar systems can detect deeper signatures, as we have pointed out in our manuscript (L340), 'It implied that at the same depth modern systems would provide not only a higher resolution, but most likely also a deeper detection of continuous IRHs (Rodriguez-Morales et al., 2020).'

We have pointed this out in the section 4.2 when we discuss the basal thermal state and 4.3.4 when we discuss the limitation of radar system. **(Revision 9)**

*Specific points:*
*Abstract line 6: probably best to use 'basal unit' to replace 'basal layer' following on Lilien et al., 2021.*

Thank you, a reasonable suggestion. But since the term "basal unit" is defined in the radargram and "stagnant ice" is what we deduce from the model, we used "stagnant ice" to replace "basal layer" here. This is also in accordance with Chung et al. (2023, in press) **(Revision 10)**

*Line 26: replace ''feasible' with 'useful'*

Done. **(Revision 11)**

*Lines 37-39:* *The Bo 2014 paper, with the 3-D model, is more of a point to the issue discussed above – they conclude that "Hence, with the observations available now, we cannot constrain the age of the basal ice well.". The 1.5 million year where the ice is not melting from their Figure 6 is an assumption used to drive their model thermodynamics, not a result.*

Thanks for pointing out the misunderstanding. We changed Line 36-39
"In the Dome A region, Sun et al. (2014) estimated ice age around Kunlun station by applying a three-dimensional, thermomechanically coupled full-Stokes model, which indicated that in the area without basal melting the ice age at 95 % depth could be limited to 1.5 Ma."
to
"In the Dome A region, Sun et al. (2014) estimated the age of ice around Kunlun station by applying a three-dimensional, thermomechanically coupled full-Stokes model assuming different geothermal flux and fabrics. They imposed a 1.5 Myr limit to the age solver, thus they did not get the actual age of the oldest ice, but the distribution of ice potentially older than maximum run time of their model." **(Revision 12)**

*Lines 40-:* *the paragraph starting at line 40 is very long and dense and could be broken up.*

Done. We splited the paragraph to three parts, the researches of the Dome Fuji ice core, the previous modelling work in the large Dome Fuji region and the topography work in the DF region. In addition, we also added the conclusions of the previous studies follow the suggestion from another reviewer. **(Revision 13)**

*Line 56:* *"on an airborne radar surveys" should be "on airborne radar surveys" or "on an airborne radar survey".*

Done. **(Revision 14)**

*Line 89:* *what is a 'two-dimension filter'?*

A 2-D filter does not just operate along a single trace (1-D), but also considers neighboring traces. In our case, it is a particular filter in the software "Echos", which we use for analysis. Since the filter could be adapted to the radar data in both trace direction and time direction, it is called a 2-D filter. This filter returns a weighted running average. It is summing up amplitudes at the same travel-time, which could remove the horizontal noise. We changed
"…a low-pass filter and a two-dimension filter are…"
to
"…a low-pass filter and a running average filter are…" **(Revision 15)**.

*Line 98:* *"in all survey lines, the third IRH H3 is" change to "in all survey lines, however the third IRH H3 is"*

Thanks for the comment, we realized that we have not expressed ourselves clearly enough in the text. We have modified Line 98
"We trace 6 or 7 relatively distinct and continuous IRHs (H1 – H7) in all survey lines, the third IRH H3 is not clear and continuous enough to be traced in some profiles"
to

"We trace 7 (H1, H2, H4 – H7) or 8 (H4 – H7) relatively distinct and continuous IRHs in the radar profiles, since the third IRH H3 is not clear and continuous enough to be traced in some profiles." **(Revision 16)**

*Line 128: "The estimate of the range precision is always higher than the resolution" - the numerical value for precision should be smaller than the resolution for a well behaved echo waveform; I would not use the term "larger" to mean "better". maybe finer verse courser?*

Thanks, we changed
"The estimate of the range precision is always higher than the resolution"
to
"The estimate of the range precision is always numerically smaller than the vertical resolution". **(Revision 17)**

*Line 264: for lazy readers who skip the methods, I would add "We show the reliability index (described in section 2.4)"*

Done. **(Revision 18)**

*Data availability: It would be good to get the IRH data at least in a repository prior to acceptance. Technical issues with getting the radar data are more understandable, but we should as a community be moving toward getting that as well. For the ice thickness product used for the modeling, would it be more appropriate to point to the Eisen et al., 2020 (https://doi.org/10.1594/PANGAEA.920234) product for the line-based data?*

We submitted the IRH data to Pangaea before submitting the manuscript, it took some time to be published and get the registered doi. The data is now available on https://doi.org/10.1594/PANGAEA.958462.

Thanks for the suggestion, after "Ice–bed returns were picked by Karlsson et al. (2018) through semi-automatic detection routines in MATLAB" we added "This ice thickness data is available on PANGAEA (Eisen et al., 2018)". **(Revision 19)**

*Figure 1: While Greene et al., 2017 should be cited if AMT was used for these plots, Greene et al is not an appropriate citation for the surface elevation data. AMT provides at least 3 different surface DEMs for Antarctica, and this paper should reference the one ultimately used.*

We have used the BedMachine plugin in AMT to plot the figure. The BedMachine data has two references, one is a data product, one is a data paper. We adjusted the citations from
"… from Greene et al. (2017) and Morlighem et al. (2017, 2020) …"
to
"… from Morlighem et al. (2020) and Morlighem (2022) ..." **(Revision 20)**

*Figure 2: what is the strong line at ~250 m depth?*

This is the radar blind zone below the surface reflection, due to saturation of the amplifier of the receive channels of the radar. The strong line is the bottom of this blind zone.

*Figure 5: gray polygons (the Van Liefferinge et al., 2018 data) on a gray scale map does not work well*

We changed the gray polygons to orange color. **(Revision 5)**

*Figure 6: the patches of blue stagnant ice are nearly invisible in this rendition. It might be better to have a separate figure or indicate existence rather than thickness. The distribution with respect to lakes you have here is interesting with comparison to Dome C where we apparently have lakes under stagnant ice.*

We think it's better to put the stagnant ice and melting map together, thus they represent reversed basal thermal states. We have tried to indicate only the existence of the stagnant ice in the figure below.

[Figure]

**Figure R3.** (a) Modelled stagnant-ice thickness and basal melt rate along the profiles of the radar survey: blue represents the existence of stagnant ice thickness and red represents the melt rate. Dark blue lines are subglacial lakes deduced from basal reflectivity in radargrams by Karlsson et al. (2018). (b) Modelled averaged accumulation rate in ice equivalent along the profiles of the radar survey.

After comparison, we felt the original design with the information of thickness works better, since the white color can represent almost no melting or stagnant ice, the blue scatters with thicker stagnant ice are the sites we care more about. To make the stagnant ice clearer, we have tripled the size of the scatters representing stagnant ice **(Revision 6)**.

We considered this comment and had a look on the lakes distribution in Dome C, we found that when there is stagnant ice modelled above the lakes, the reliability of the model results is relatively lower. Also, sometimes there is basal melting underneath the basal unit, as the basal unit is advected from regions of thinner ice.

***Figure 7:*** *it's very hard to tell what is going on with the overlapping color zones. Especially if one is color blind - STD and Run III could look identical.*

We have tried different color groups and finally used blue, dark red and orange to replace the black, blue and red to make figure more visible. However, the uncertainties of RUN II and RUN III are similar, so their shades are overlapped mostly, which makes it hard to tell them apart. We have pointed this out in the caption. In addition, we have changed the size of marks of isochrones, and added more information, e.g., the depth and age of the 7[th] and 8[th] isochrones, the location of the inflection point. **(Revision 21)**

**References**

Chung, A., Parrenin, F., Steinhage, D., Mulvaney, R., Martín, C., Cavitte, M. G. P., Lilien, D. A., Helm, V., Taylor, D., Gogineni, P., Ritz, C., Frezzotti, M., O'Neill, C., Miller, H., Dahl-Jensen, D., and Eisen, O. (2023): Stagnant ice and age modelling in the Dome C region, Antarctica, EGUsphere, 2023, 1–31.

Drews, R., Eisen, O., Weikusat, I., Kipfstuhl, S., Lambrecht, A., Steinhage, D., Wilhelms, F. and Miller, H. (2009): Layer disturbances and the radio-echo free zone in ice sheets, The Cryosphere, 3, pp. 195-203.

Franke, S., Gerber, T., Warren, C., Jansen, D., Eisen, O., & Dahl-Jensen, D. (2023). Investigating the radar response of englacial debris entrained basal ice units in East Antarctica using electromagnetic forward modelling. IEEE Transactions on Geoscience and Remote Sensing.

Obase, T., Abe-Ouchi, A., Saito, F., Tsutaki, S., Fujita, S., Kawamura, K., and Motoyama, H. (2022): A one-dimensional temperature and age modeling study for selecting the drill site of the oldest ice core around Dome Fuji, Antarctica, The Cryosphere Discussions, 2022, 1–24.

Rodriguez-Morales, F., Braaten, D., Mai, H. T., Paden, J., Gogineni, P., Yan, J.-B., Abe-Ouchi, A., Fujita, S., Kawamura, K., Tsutaki, S., et al. (2020): A Mobile, Multichannel, UWB Radar for Potential Ice Core Drill Site Identification in East Antarctica: Development and First Results, IEEE Journal of Selected Topics in Applied Earth Observations and Remote Sensing, 13, 4836-4847.

**Response to RC2 of 'Mapping age and basal conditions of ice in the Dome Fuji region, Antarctica, by combining radar internal layer stratigraphy and flow modeling' (tc-2023-35)**

We thank Anonymous Referee #2 for these very helpful and thoughtful review comments. We are most grateful for the time the reviewer spent providing feedbacks on how to improve our manuscript. In our revision, we have tried to address the reviewer's suggestions as much as possible, specified in detail below. For clear tracking of changes, individual issues raised by the referees are referred to as (**Revision I**) below, where "I" is the number of the comment in the attached table. The line number of each revision could be found in the table.

*This manuscript seeks to map out properties of the ice near the bed around Dome Fuji, East Antarctica. The authors trace layers in an airborne radar survey from 2017/2018, and date those layers using the age scale from an ice core drilled at Dome Fuji proper. They then fit a 1D pseudo-steady model (that has been applied extensively to Dome C) to the isochrones at each trace in the radar survey, from which they get an average accumulation, shape factor, and effective ice thickness (indicating stagnant ice or basal melting) at that point. The results suggest very old ice in the area, albeit with enormous (and, according to the authors, underestimated) uncertainty.*
*In the end, the conclusions here are rather thin. The issue with taking the method of Chung et al., in review, and using it on this survey is that the isochrones used here are much less than half as old (170 ka vs 476 ka), and the resultant impact on the reliability of the results is enormous. Essentially, the issue is that a 170 ka isochrone tells us very little about 1.5 Ma ice. Multiple problems can occur: small violations in model assumptions will result in inferred ages that are unrelated to reality (e.g. if ice flow, ice thickness, or accumulation varied in unexpected ways), and overfitting to measurement errors on these young isochrones will cause incorrect results at older ages. Indeed, when the authors check this possibility with even a 230 ka isochrone, they find that there is a huge change in ages as we would expect if the young isochrones simply do not carry much information about deeper ages. In my view, that this problem is occurring is demonstrated conclusively since the model does not match the Dome Fuji ice core's age scale. This is the only really available test of the model reliability—and not only does the model miss the age scale, but it does so outside its reported uncertainty! That is to say, the model is both wrong and confidently so. As a result, I do not think that much can be concluded from this paper, other than that old ice may or may not exist at Dome F. To their credit, the authors describe the limitations in the discussion and conclusion (though it should be better disclosed in the results), so I think that after revision the work will be publishable in The Crysosphere.*
*I think the presentation could be improved in both the paragraph-scale structure and at the sentence level, and the figures could be improved as well. The paper is in need of quite a bit of grammatical work, as there are a lot of missing or extra articles and some subject/verb mismatch. I did not enumerate these in my review. At times I found the paper hard to read, although I think I could eventually discern the meaning, so this could probably be handled by a copy editor. At some points, paragraphs wander away from the thread of the manuscript (see general comments). I agree with the other reviewer that the figures are pretty difficult to make out given their size. Throughout the manuscript, colorbars start and end at arbitrary values and demarcations are difficult to read—well chosen start and end values, arrows to indicate whether the colorbar values are inclusive, and enlargement would help.*

To avoid the misunderstanding of the dataset we have used in this study, it is important to clarify that the radar dataset was collected with the AWI radio-echo sounding (RES) system

during 2016-2017 Antarctica season, not the dataset collected with the UWB radar in 2017-2018 season. As stated in the system description, this is a pulse system which records rectified waveforms, i.e., it does not record any phase information, thus yielding lower resolution, especially of internal layers.

Because of the data quality, the continuous internal reflection horizons (IRHs) can only be traced in the upper part and dated back to 170 ka. This limitation could not be solved from the data side. We agree with the comment that the unexpected variations of thinning in the deep part could lead to the mismatch between the model and reality. As the reviewer pointed out, there is a significant overestimation of age of basal ice at DF (Figure 7), however, we think that this figure actually proved the reliability of the model to a certain extent.

In Figure 7, we show the age-depth scale derived from models at Dome Fuji. We pointed out the age of basal ice was overestimated by a factor of 2 at DF. This figure shows, for RUN II (the run with 8 IRHs), that the deepest IRH was traces at the depth of 2238.22 m at DF, and dated back to 232.65 ka BP. We consider that the model has a reasonable performance (agreement with the timescale DFO2006+AICC2012) to the depth of 2759 m, which is dated back to 540.48 ka BP. This depth is about 290 m above the bed, which is exactly the depth of the inflection point in the timescale DFO2006+AICC2012. As a model assumption, the age-depth profile of the model follows the exponential distribution below this depth. Thus, the large overestimation is actually caused by the curvature reversal below this depth in the timescale of the ice core.

A similar phenomenon was also observed in our companion study (Chung et al. in press) at EDC though much older IRHs were dated there. They pointed out: "The modelled age at the deepest dated point for the EDC drill site which was around 100-200 kyr older than would be expected from the AICC2012 age-depth profile ... Looking at the AICC2012 profile determined by experimental measurements, it follows an exponential profile until the lower 200 m of dated ice, perhaps meaning that the thinning is for some reason lower than the model would expect…".

Figure R4 here simply shows the comparison between the age-depth profile derived by the model and the timescale at EDC (AICC2012). The model result agrees with the timescale perfectly until the timescale deviates from the exponential form. At DC the overestimation is more reasonable since the timescale at DC doesn't change as drastically as that at DF. In the DF case, to solve this overestimation problem, only more continuous isochrones below the inflection point, i.e., the lowest 300 m, could provide better constrains for the model, which is not possible in our dataset, and in fact most likely also not easily possible with other data sets (Tsutaki et al., 2022).

The analysis of the DF deep ice core indicates melting at DF (Motoyama et al., 2021), which could explain the inflection point in the age-depth scale of the ice core. This would imply that the significant overestimation likely occurs in the area with basal melting. Given that there is only one deep ice core in the DF region, we lack an additional timescale extending towards the bedrock to prove our hypothesis. At the same time, we find it unjustified to draw the conclusion that the model approach does not work in the DF region. According to what we have found by now, the model works quite well in the upper 2/3 of the ice column by calculating our reliability index (the standard deviation of the age difference between observation and model results), and it also seems appropriate in the deeper part at DF, until reaching the depth of the inflection point of timescale.

Therefore, we proposed "the reliability of our model is probably overestimated" in the manuscript as a responsible statement. In fact, this probability is higher in the area with basal melting. We highlighted this hypothesis and emphasize the importance of considering basal thermal state while finding old ice by our model more in the revision. Since we have found that melting prevails over stagnant ice in the DF region by using our model approach, it is possible that significant overestimations of age occur in the deep ice in the ice at various places, as what has been observed at DF. Taking this into consideration, we included an additional Figure 4d in the manuscript that depicts the age of ice at a depth of 300 m above the bed. This figure could provide relatively accurate age values while excluding the lowest part of the ice. **(Revision 22)**

[Figure]

**Figure R4.** Model derived age-depth scale and AICC2012 timescale at EDC.

In the revised version of the manuscript, we improved the section 4.3.3 in the sense of what we have discussed here. **(Revision 23)**

As the reviewer suggested, we also improved our conclusion. **(Revision 24)**

We followed the comments and refine our figures by adding the start and end values to the color bars and placing the ticks outside the color bars. **(Revision 3-6, 25)**

We conducted a thorough review of the manuscript, carefully addressing and correcting the grammar and vocabulary errors. **(Revision 26)**

*General comments*
*Basal layer is a sticky term. As used on line 26, it sounds like generic deep ice. However, in other literature, it refers specifically to ice that has a distinct radar character, to the ice near the bottom of EDC that has little discernible paleoclimatic information, or to ice that is inferred to be stagnant. The situation is further muddled on line 70, where the authors suggest that their method can detect a potentially stagnant basal layer—but this is incorrect. The method can only detect a stagnant basal layer (at least in terms of vertical velocity), since the whole premise of the detection is that the basal layer is stationary for the purposes of the depth-age scale. This work should be consistent on its usage of the term basal layer and it should define what it means by that in the introduction. Imprecise usages such as that in line 26 should be removed. Some discussion should be added on whether there is any correspondence between the areas where stagnant ice is inferred and any characteristic of the radargrams.*

Thanks for pointing out the confusing use of 'basal layer'. We decided to use 'basal unit' as the bottommost part (which has been used in the literature before) in which there are some peaks in return power but no continuous or coherent reflecting horizons (and in consistency with other recent publications). We used "stagnant ice" as the stagnant bottommost ice above the ice-bed interface derived from the model.

We changed "basal layer" to "deepest ice" in Line 26. **(Revision 27)**

We changed "a (potentially stagnant) basal layer" to "the stagnant ice" in Line 70. **(Revision 28)**

Except for the two changes suggester by reviewer, we also changed "...there is a basal layer of stagnant ice…" to "...there is stagnant ice…" in Line 153. **(Revision 29)**

We changed "basal layer" to "the lowermost ice" in Line 367. **(Revision 30)**

In the radar dataset we used, there is an echo free zone (EFZ) above the bed, with a thickness of several hundreds of meters. EFZ could be caused by various reasons, e.g., system sensitivity of the radar system, deformation, recirculation and recrystallization of ice (Drews et al., 2009 and Franke et al., 2023). We think the EFZ in our dataset could most likely be caused by the performance of the radar system, since in the same region more modern radar can detect somewhat deeper, more coherent horizons, as we have pointed out in our manuscript (L340), 'It implied that at the same depth modern systems would provide not only a higher resolution, but most likely also a deeper detection of continuous IRHs (Rodriguez-Morales et al., 2020).' We pointed this out in section 4.2 when we discuss the basal thermal state and 4.3.4 when we discuss the limitation of radar system. **(Revision 9)**

*Given the wide pulses of this radar system, and the lack of pulse decompression, I am skeptical of the vertical precision that the authors are claiming and I would suspect a bias. Just by two way travel time, the pulse is 50 m long (as the authors correctly point out, this is the resolution, defined here as the separation needed to identify two targets as distinct). Again, as the authors correctly identify, this resolution is different than the precision (i.e. the depth-accuracy of a target). However, the authors trace in a fairly standard manner (picking*

*the strongest return), but given the processing of these data the depth of the reflector should really be off the first return, assuming that time zero is defined as the start of the pulse (indeed, this is what you would get if you could do pulse decompression). The problem is that I would expect the strongest return to lie below the first return from a reflector (most likely 25 m below, but this offset is somewhat arbitrary), but never above. Thus, I think that there is likely a systematic bias in the ages and depths used. This may have a small effect on age-depth scales in the end, since it may affect an isochrone in the same manner along its length, but it should be accounted for carefully.*

To better illustrate the performance of the system and underline our uncertainty estimates, in the next figures R5 and R6 we show some screenshots of the radar returns of different IRHs, the horizontal axis shows traces, the vertical axis shows Time ($\mu s$) (the unit is $s$ on the figure, because the software is originally used for seismic data).

[Figure]

**Figure R5.** Zoom-in view of IRH4 returns in Profile 20172029.

[Figure]

**Figure R6.** Zoom-in view of IRH7 returns in Profile 20172029.

For a single pulse, the bias between the first return and the strongest return varies, but generally ranges from 50-200 ns.

For IRH7, in most traces, multiple return pulses overlap. E.g., for trace 22815, the first return occurs at 33430 ns, but the strongest return we traced could be at 33850 ns, which results in an offset of around 420 ns. The bias varies from 50-420 ns in the best case, which is too big to be a systematic bias regarding the pulse length of 600 ns.

Hence, we prefer to keep our revised description for the age uncertainty. **(Revision 8)**

*Overall, uncertainty deserves a more prominent place in the results and discussion. First, how is the basal age uncertainty that the authors report calculated? I am guessing it is as in Chung et al., but there is not even a passing mention in this manuscript of how the authors obtain anything other than a best-fit value—it is critical to add this to the manuscript. Then, there is the issue of uncertainty in basal age ice—the number reported for Dome F itself is plus minus 500 kyr. This is enormous, over a third of the age. Figure 4 should plot the uncertainty on the basal ages—without this, the reader has no idea if there are any areas at all with reliably old ages. This gets addressed later by table 1 and the sensitivity analysis, but I see no reason that it cannot fit in Figure 4.*

Thanks for pointing this out. The theory is the same as the one described in Chung et al. (2023, in press), which provides a detailed account of the approach. Nevertheless, we added more details of the age uncertainty at the end of the Method session to make it clearer. **(Revision 1)**

The value of the uncertainty of the basal age is a significant quantity in the entire DF region. We relate this phenomenon to the fact that the number and depth of IRHs used as constraints for the model are limited by the traceability of the IRHs in our radar dataset. IRHs constraints help with determining the shape of the thinning function (p factor), so of course having more IRHs will give us a more accurate p value. The thinning function is almost linear in the upper section of the ice sheet and then becomes non-linear in the deepest part. Since the IRHs we have traced are at the top 2/3 of the ice thickness, we can not constrain the model well in the lower 1/3. However, this lower section has the largest impact on the p value.

We have compared it with that from the Dome C region (Chung et al., 2023). The age uncertainty of each isochrone is similar in Dome C, but the basal age uncertainty is much smaller. This is likely due to more IRH constraints covering a larger portion of the ice sheet thickness in the DC region. We added the age uncertainty of the basal ice as Figure 4c. **(Revision 31)**. In fact, we consider this comparison important, as the same approach applied in different regions and/or different radar data can yield a quite different uncertainty.

*The comparison with Karlsson 2018 and van Liefferinge 2018 deserves more consideration. While the approaches are different, what can we learn by comparing them? Where should we believe their results over the results here (for example, are the results here thermodynamically tenable)? Where is there agreement?*

Karlsson et al. (2018) and van Liefferringe et al., (2018) used a thermodynamical model to identify old ice sites, but their approaches did not include any constraint from the age of radar internal layers. The model we are using does not take into account the thermal dynamics at all, thus being more independent of GHF estimates. However, the sites with potentially "old ice" suggested by the different approaches show a considerable correspondence in some places, especially around DF and NDF. We consider that the two main underlying reason for this consistency is the use of ice thickness in both models, which has implied the important

impact of ice thickness on the age distribution of the ice, as well as the validity of the approximations regarding the thinning function in our approach.

In the figure, we put the sites with old ice suggested by the thermal model on top of our results. In this way we aim at giving readers a visual impression, which site is more likely to hold old ice, both mechanically and thermodynamically. We have rephrased the paragraph to make it more logical. **(Revision 32)**

*The introduction could use some work. We would benefit from more focus on what the reader should take away. This is most obvious in the paragraph beginning at line 40 wanders between detailed analysis of timescales, studies that concluded something about basal thermal state and potential old ice sites (without ever stating those conclusions). I suggest a careful culling of the introduction, focusing on the goals of each paragraph.*

We splited the paragraph to three parts, the researches of the Dome Fuji ice core, the previous modelling work in the large Dome Fuji region and the topography work in the DF region. In addition, we also added the conclusions of the previous studies follow the suggestion from another reviewer. **(Revision 13)**

*Detailed comments*

*L36: If efforts beyond BE-OI are discussed, it seems strange to include Dome A but not other countries' oldest ice efforts. For example, Beem et al. (2021) would then also be appropriate here. Also on L36, the sentence needs to be rephrased—is the point that the age is limited or that it reaches 1.5 Ma?*

We added the reference as suggested to have a complete overview. **(Revision 33)**

We changed Line 36-39 "In the Dome A region, Sun et al. (2014) estimated ice age around Kunlun station by applying a three-dimensional, thermomechanically coupled full-Stokes model, which indicated that in the area without basal melting the ice age at 95 % depth could be limited to 1.5 Ma."
to
"In the Dome A region, Sun et al. (2014) estimated the age of ice around Kunlun station by applying a three-dimensional, thermomechanically coupled full-Stokes model assuming different geothermal flux and fabrics. They imposed a 1.5 Myr limit to the age solver, thus they did not get the actual age of the oldest ice, but the distribution of ice potentially older than maximum run time of their model." **(Revision 12)**

*L81: I find this system description to be a bit vague. I was under the impression that the AWI system is multi-channel and phase-coherent. Is this a single channel power, or the total transmit power? Pulse-limited is not a term we see that often for ice-penetrating radars (perhaps more for radar altimeters), so it would be helpful to say the importance explicitly— that it acts much like a chirp system where you cannot decompress the chirps (thus the very low vertical resolution). Is there no reported bandwidth because there was no frequency sweep in the chirp? If the chirp did sweep frequencies, the bandwidth should be reported.*

The dataset we have used in this research was collected with the AWI radio-echo sounding (RES) system during the 2016-2017 Antarctica field season, not the dataset collected with the AWI UWB radar in the 2017-2018 season.

The AWI RES system is a burst system, with center frequency of 150 MHz. The pulse generated has length of 60 ns or 600 ns, as we referenced in our manuscript (Nixdorf et al., 1999). The signal transmitted by the system is not chirp with a frequency sweep, but a short, high-power pulses with specific widths, which is in addition rectified after reception. We also mentioned the bandwidth in Line 82. We changed
"The AWI RES system transmits radar waves with a center frequency of 150 MHz and an amplitude of 1.6 kW."
to
"The AWI RES system transmits radar waves with a center frequency of 150 MHz, a band width of 20 MHz and an amplitude of 1.6 kW." **(Revision 34)**

*L89: Should explicitly state what kind of 2d filter*

It is a particular filter in the software "Echos", which we use for analysis. Since the filter could be adapted to the radar data in both trace direction and time direction, it is called a 2D filter. This filter returns a weighted running average of neighboring traces. It is summing up amplitudes at the same travel-time, which could remove the horizontal noise. We changed
"…a low-pass filter and a two-dimension filter are…"
to
"…a low-pass filter and a running average filter are…" **(Revision 15)**.

*Figure 1: "examplary" implies that it is an ideal or "the best" profile, while I think simply "example" would be more accurate. Different colors for contours and survey would help readabilility. An increase in size would be nice too.*

We changed "exemplary" to "example". **(Revision 20)**

We changed the color of the contours. **(Revision 3)**

*Figure 2: The horizontal black lines make this almost impossible to interpret. I would also like to see a zoom in on some of the tracing, so that we can see how the traced depth relates to the width of the returns. Why not just use a plus/minus for the uncertainty and avoid the confusing double parenthetical?*

We removed the black lines and improve the figure.

[Figure]

**Figure R7.** Zoom-in view of IRH4 picked in Profile 20172040.

We show a zoom-in view of profile 20172040 in Fig. R7. It is the same as the returns in zoom-in view of different horizons in Profile 20172029, which we have shown in General comment. This illustrates that there is no trustworthy systematic bias that could be adapted to all the picks.

Thanks for the comment, we used plus/minus to replace double parenthetical. **(Revision 35)**

*L140: This needs to state how accumulation is inferred for ages that predate the oldest ice in the core*

Thanks, we stated how accumulation rate was inferred. We added 'We assume r(t) = 1 beyond the extent of the DF ice core record (715~ka).' **(Revision 36)**

*L141: "Inverted" means "was turned upside down"—but this was both inverted and integrated. Here "inferred" or simply "calculated" would be correct.*

Thanks, we changed "inverted" to "inferred". **(Revision 37)**
However, we want to emphasize that it is a general problem that some scientist use "inverted for …" and other do not consider this a proper term. We had dealt with a comparable comment already 2007 from Ed Waddington, but the community has not come up with a reasonable agreement on how to deal with such an ambiguity in usage.

*L142: This needs to be rephrased to make clear that the presence of stagnant ice is undetermined*

Thanks for the comment. From Line 147, we reorganized the paragraphs, by moving the description of $H_m$ forward, and then described the optimization.

We changed
"…where $H_m$ is the mechanical ice thickness, which means the effective ice thickness above the stagnant ice, and $p$ is a shape factor controlling vertical deformation (Lilien et al., 2021)."
to
"…where $p$ is a shape factor controlling vertical deformation (Lilien et al., 2021), $H_m$ is the mechanical ice thickness, which is different to the observed ice thickness $H_{obs}$. When $H_m$ is greater than the observed ice thickness $H_{obs}$, we have melting conditions at the base. Otherwise, there is stagnant ice. If the basal ice is melting, the melt rate $m$ can be obtained by…" **(Revision 38)**

*L149: Considering that Chung et al. And Lilien et al. use inverse methods that produce uncertainties, it is important to state here whether this method does the same or whether this work simply finds the single solution with the lowest misfit.*

We added a few paragraphs to describe the optimization methods and rephrased the method section. **(Revision 1)**

*L156: Could this be renamed? It is essentially the inverse of the reliability, so it is quite confusing to call it the reliability index.*

For consistence with our companion manuscript, we prefer to keep the same naming. But to avoid misunderstanding, we added "$\sigma_R$" after "reliability index". **(Revision 39)**

*3.1: Same issue with meaning of "exemplary"*

We changed "exemplary" to "example". **(Revision 40)**

*L171: Delete parenthetical—thickness cannot be deep*

Done. **(Revision 41)**

*Figure 5: The Van Liefferinge data need a different color—gray on a gray background is unreadable.*

We changed the gray polygons to orange color. **(Revision 5)**

*Final paragraph: I do not think this is a conclusion, nor is it a logical way to approach the limitations of this work. Overall, it is a rather weak note to end on—why not move it to the discussion, which is what it really is anyway?*

We rewrote the conclusion. **(Revision 25)**

**References**

Chung, A., Parrenin, F., Steinhage, D., Mulvaney, R., Martín, C., Cavitte, M. G. P., Lilien, D. A., Helm, V., Taylor, D., Gogineni, P., Ritz, C., Frezzotti, M., O'Neill, C., Miller, H., Dahl-Jensen, D., and Eisen, O. (2023): Stagnant ice and age modelling in the Dome C region, Antarctica, EGUsphere, 2023, 1–31.

Drews, R., Eisen, O., Weikusat, I., Kipfstuhl, S., Lambrecht, A., Steinhage, D., Wilhelms, F. and Miller, H. (2009): Layer disturbances and the radio-echo free zone in ice sheets, The Cryosphere, 3, pp. 195-203.

Franke, S., Gerber, T., Warren, C., Jansen, D., Eisen, O., & Dahl-Jensen, D. (2023). Investigating the radar response of englacial debris entrained basal ice units in East Antarctica using electromagnetic forward modelling. IEEE Transactions on Geoscience and Remote Sensing.

Karlsson, N. B., Binder, T., Eagles, G., Helm, V., Pattyn, F., Van Liefferinge, B., and Eisen, O. (2018): Glaciological characteristics in the Dome Fuji region and new assessment for "Oldest Ice", The Cryosphere, 12, 2413–2424.

Lilien, D. A., Steinhage, D., Taylor, D., Parrenin, F., Ritz, C., Mulvaney, R., Martín, C., Yan, J.-B., O'Neill, C., Frezzotti, M., Miller, H., Gogineni, P., Dahl-Jensen, D., and Eisen, O. (2021): Brief communication: New radar constraints support presence of ice older than 1.5 Myr at Little Dome C, The Cryosphere, 15, 1881–1888.

Motoyama, H., Takahashi, A., Tanaka, Y., Shinbori, K., Miyahara, M., Yoshimoto, T., Fujii, Y., Furusaki, A., Azuma, N., Ozawa, Y., et al. (2021): Deep ice core drilling to a depth of 3035.22 m at Dome Fuji, Antarctica in 2001–07, Annals of Glaciology, 62, 212–222.

Nixdorf, U., Steinhage, D., Meyer, U., Hempel, L., Jenett, M., Wachs, P., and Miller, H. (1999): The newly developed airborne radio-echo sounding system of the AWI as a glaciological tool, Annals of Glaciology, 29, 231–238.

Parrenin F, Bazin L, Capron E, et al. IceChrono1(2015): A probabilistic model to compute a common and optimal chronology for several ice cores, Geoscientific Model Development, 8(5): 1473-1492.

Rodriguez-Morales, F., Braaten, D., Mai, H. T., Paden, J., Gogineni, P., Yan, J.-B., Abe-Ouchi, A., Fujita, S., Kawamura, K., Tsutaki, S., et al. (2020): A Mobile, Multichannel, UWB Radar for Potential Ice Core Drill Site Identification in East Antarctica: Development and First Results, IEEE Journal of Selected Topics in Applied Earth Observations and Remote Sensing, 13, 4836-4847.

Tsutaki, S., Fujita, S., Kawamura, K., Abe-Ouchi, A., Fukui, K., Motoyama, H., Hoshina, Y., Nakazawa, F., Obase, T., Ohno, H., Oyabu, I., Saito, F., Sugiura, K., and Suzuki, T. (2022): High-resolution subglacial topography around Dome Fuji, Antarctica, based on ground-based radar surveys over 30 years, The Cryosphere, 16, 2967–2983.

Van Liefferinge, B., Pattyn, F., Cavitte, M. G., Karlsson, N. B., Young, D. A., Sutter, J., and Eisen, O. (2018): Promising Oldest Ice sites in East Antarctica based on thermodynamical modelling, The Cryosphere, 12, 2773–2787.

| Revision number | Reviewer | Position | Position after revision | Before | Revision | | Note |
|---|---|---|---|---|---|---|---|
| 1 | 1 2 | Section 2.4 | Section 2.4 | | 1)The final revised Chung et al. (2023) paper is published online now. 2)Add further details of the model. | | |
| 2 | 1 | Line 211 | Line 255, 265, 290, 410 | | 1)The final revised Obase et al. (2023) paper is published online now. 2)Rephrase the comparison in our manuscript regarding their revisions of model results. | | |
| 3 | 1 2 | Figure 1 | Figure 1 | | | Change the contours color of the surface elevation. | |
| 4 | 1 2 | Figure 4 | Figure 4 | | | We add Figure 4c and 4d, see Revision 22 and 31. | We replace the gray scale bed elevation background with colored contours with different line style and show it in the response letter. |
| 5 | 1 2 | Figure 5 | Figure 5 | | 1)We increase the size of data points in Figure 1,4,5,6. 2)We increase the size of markers of DF and NDF in the figures. 3)We give the values of colorbars at start and end. 4)We put the ticks outside the color bar to see them clearly. | Add ellipses to show the old ice sites. Change gray polygon to orange. | |
| 6 | 1 2 | Figure 6 | Figure 6 | | | Specially enlarge the scatters of stagnant ice | We plot two figures, the first one we enlarge the scatters, the second one we use light blue for all the stagnant ice, we show it in the response letter. We think it's better to use the first figure, since when there is white, means there is no melting or very shallow stagnant ice, which we would not consider as potential for old ice sites. We care more about thicker stagnant ice in darker blue color. |
| 7 | 1 | Section 4.3.1 | Section 4.3.1 | | Rephrase this part and provide velocity map in Fig. 1b. | | |
| 8 | 1 2 | Section 2.3 | Line 139 | | We rephrase the age uncertainty part in a clearer way in revision. | | |
| 9 | 1 2 | Section 4.2/4.3.4 | Line 308, 459, 496 | | We point out the limitation of radar system, i.e., basal unit couldn't be observed in the radargrams. | | |
| 10 | 1 | Line 6 | Line 7 | basal layer | stagnant ice | | |
| 11 | 1 | Line 26 | Line 26 | feasible | useful | | |
| 12 | 1 2 | Line 37-39 | Line 37-39 | In the Dome A region, Sun et al. (2014) estimated ice age around Kunlun station by applying a three dimensional, thermomechanically coupled full-Stokes model, which indicated that in the area without basal melting the ice age at 95 % depth could be limited to 1.5 Ma. | In the Dome A region, Sun et al. (2014) estimated ice age around Kunlun station by applying a three-dimensional, thermomechanically coupled full-Stokes model assuming different geothermal flux and fabrics. They imposed a 1.5 Myr limit to the age solver, thus they didn't get the actual age of the oldest ice, but the distribution of ice potentially older than maximum run time. | | |
| 13 | 1 2 | Line 56- | Line 44 | | 1)Split the paragraph to three parts, the researches of the Dome Fuji ice core, the previous modelling work in the large Dome Fuji region and the topography work in the DF region. 2)Add the conclusions of the previous studies follow the suggestion from another reviewer. | | |
| 14 | 1 | Line 56 | Line 62 | …based on an airborne radar surveys… | …based on airborne radar surveys… | | |
| 15 | 1 2 | Line 89 | Line 98 | …two–dimension filter… | …running average filter… | | |
| 16 | 1 | Line 98 | Line 106 | We trace 6 or 7 relatively distinct and continuous IRHs (H1--H7) in all survey lines, the third IRH H3 is not clear and continuous enough to be traced in some profiles. | We trace 6 (H1, H2, H4 – H7) or 7 (H1 – H7) relatively distinct and continuous IRHs in the radar profiles, since the third IRH H3 is not clear and continuous enough to be traced in some profiles. | | |
| 17 | 1 | Line 128 | Line 139 | The estimate of the range precision is always higher than the resolution. | The estimate of the range precision is always numerically smaller than the vertical resolution… | | |
| 18 | 1 | Line 264 | Line 325 | …We show the reliability index in the… | … We show the reliability index (described in section 2.4) in the… | | |
| 19 | 1 | Line 99 | Line 109 | | Add "This ice thickness data is available on PANGEA (Eisen et al., 2020)." after "…through semi-automatic detection routines in Matlab…" | | |
| 20 | 1 2 | Figure 1 caption | Figure 1 caption | (1)…from Greene et al. (2017) and Morlighem et al. (2017, 2020) (2)…examplary… | (1)…from Morlighem et al. (2020) and Morlighem (2022) (2)…example… | | |
| 21 | 1 | Figure 7 | Figure 7 | | 1)Use blue, dark red and orange colour for STD, RUN II and RUN III. Because of the overlap between uncertainty shadows of RUN II and RUN III, it's a little bit hard to tell them apart. The differences between STD and RUN II/III is clear, these are what we care about. We add 'we note that the uncertainties of RUN II and RUN III are similar, so their shades are overlapped mostly.' in the caption of Figure 7. 2) Make the markers size bigger. 3)Add inflection point of the timescale DFO2006+AICC2012. 4)Add the age and depth of EH8 and inflection point at DF. 5)Add line depth = 2350 m. | | |
| 22 | 2 | | Figure 4d | | Add age of ice at the depth of 300 m above the bed as Figure 4d. | | |
| 23 | 2 | Section 4.3.3 | Section 4.3.3 | | Rewrite this section. | | |
| 24 | 2 | Conclusion | Conclusion | | Refine the conclusion. | | |
| 25 | 2 | Figure 3 | Figure 3 | | We put the ticks outside the color bar to see them clearly. | | |
| 26 | 2 | Grammer | | | Revise carefully. | | |
| 27 | 2 | Line 26 | Line 26 | basal layer | deepest ice | | |
| 28 | 2 | Line 70 | Line 79 | a (potentially stagnant) basal layer | the stagnant ice | | |
| 29 | 2 | Line 153 | Line 159 | ..there is a basal layer of stagnant ice… | …there is stagnant ice… | | |
| 30 | 2 | Line 367 | Line 505 | basal layer | the lowermost ice | | |
| 31 | 2 | | Figure 4c | | Add age uncertainty of basal ice as Figure 4c. | | |
| 32 | 2 | Line 224 | Line 268 | The two previous studies…although the approaches are very different. | The two previous studies by Karlsson et al. (2018) and Van Liefferinge et al. (2018) are based on a thermodynamic model, considering regions with an surface ice flow velocity smaller than 1 m a−1. Their main constraint for the presence of old ice 270 is that the GHF is not sufficiently large to cause temperate conditions at the base, and thus melting. Another criterion is ice thicker than 2000 and 2500 m, respectively. They suggested several potential areas holding old ice, which are displayed by semi-transparent pink and orange shades in Fig. 5, respectively. Our approach, in contrast, is solely based on the observed age– depth distribution, which is then extrapolated to larger depth by using observed accumulation rates and making assumptions about the thinning function. | | |
| 33 | 2 | | Line 41 | | Add ' Beem et al. (2021) suggested that Titan Dome is an unlikely potential site for drilling an old ice core covering MPT based on depth of dated internal layers, age models and ice flow velocity.' | | |
| 34 | 2 | Line 82 | Line 91 | The AWI RES system transmits radar waves with a center frequency of 150 MHz and an amplitude of 1.6 kW. | The AWI RES system transmits radar waves with a center frequency of 150 MHz, a band width of 20 MHz and an amplitude of 1.6 kW. | | |
| 35 | 2 | Figure 2 | | | (1)Remove the black lines. (2)Use plus/minus to replace parenthesis. | | |
| 36 | 2 | Line 140 | Line 161 | | Add 'We assume r(t) = 1 beyond the extent of the DF ice core record (715–ka). ' | | |
| 37 | 2 | Line 141 | Line 150 | inverted | inferred | | |
| 38 | 2 | Line 147 | Line 170 | …where $H_m$ is the mechanical ice thickness, which means the effective ice thickness above the stagnant ice, and p is a shape factor controlling vertical deformation (Lilien et al., 2021)… | …where p is a shape factor controlling vertical deformation (Lilien et al., 2021), $H_m$ is the mechanical ice thickness, which is different to the observed ice thickness $H_{obs}$...When $H_m$ is greater than the observed ice thickness $H_{obs}$, we have melting conditions at the base. Otherwise, there is stagnant ice. If the basal ice is melting, the melt rate m can be obtained by… | | |
| 39 | 2 | Line157/Fig 5 | Line 184, Fig 5, Line 325 | reliability index | reliability index $\sigma R$ | | |
| 40 | 2 | Line167/168 | Line 194/195 | exemplary | example | | |
| 41 | 2 | Line171 | Line 173 | ...where the mechanical ice thickness Hm (purple dash line) is larger (deeper) than the observed ice thickness (black line)… | ...where the mechanical ice thickness Hm is larger  than the observed ice thickness… | | |
| 43 | | All the orientations | | | All directions should be referring to true north, what we used before were actually grid orientations in polar  stereographic coordinates, thus, we change all the orientations. | | |
| 44 | | Data avalibility | | The ice thickness observed from radar data are published on https://doi.org/10.1594/PANGAEA.891323 (Karlsson et al., 2018). | The ice thickness observed from radar data are published on https://doi.org/10.1594/PANGAEA.920234 (Eisen et al., 2020). | | |
| 45 | | Data availability | | We are waiting for the raw radar data and IRHs traced in radargrams being available on data repository | The IRH data are available through the Pangaea repository https://doi.org/10.1594/PANGAEA.958462 (Wang et al., 2023) | | |
| 46 | | Update the model results | | | Update new model results (figures and values) since the model was slightly improved. | | |

---

## Author Response (AR2)

Response to editor of 'Mapping age and basal conditions of ice in the Dome Fuji region, Antarctica, by combining radar internal layer stratigraphy and flow modeling' (tc-2023-35)

August 7, 2023

*The manuscript is improved considerably after the first round of revisions. I would ask that the authors make a careful check of the text for English usage and grammar, and also look at my comments below.*

We thank the editor Benjamin Smith for offering helpful review comments to improve the quality of our manuscript. We addressed suggestions specified in detail below.

*Throughout the manuscript:*
*--Model results are presented with far too many significant digits for the uncertainty in the estimates. For example : an age of 1933.7±769.3. To follow rules for significant digits, this should be 1900 ± 800, but common practice allows 1930 ±770. Please check your significant digits, and do not provide more than one order of magnitude precision beyond the estimated precision of the measurements.*

Thanks for the comment, we have changed the precision of modelled values in our manuscript. E.g., we changed modelled age from 1933.7±769.3 ka to 1930 ±770 ka, we changed thickness of stagnant ice from 28.5 m to 29 m, we changed basal melt rate from 1.36 mm a$^{-1}$ to 1.4 mm a$^{-1}$. And we corrected the corresponding figures.

*--The authors use the word "corresponds" to mean something like "agrees with." More conventionally, this word is used to mean "plays the same role as" and does not specify whether the two quantities compared agree or disagree. Please check each use of this word in the manuscript, and consider what it is intended to mean.*

Thanks for the comment, we have checked all the "correspond" and change some to "agree with" when we want to specify if the two quantities compared agree or not.

*120: double "the"*

We deleted the repetitive "the".

*285: regionally-> regional*

We changed "regionally" to "regional".

*288: "i.e. in a larger region" – this parenthetical comment is not clear. Please explain or delete.*

We deleted the comment in the parentheses.

*294: "corresponds" is too vague. Please find a specific word for the relationship between these studies*

We changed "corresponds to" to "agrees with".

*316, 327, 378: delete "relatively"*

We deleted "relatively" in Line 316, 327, 338.

*330-348: This section is not clearly relevant to the topic heading "model reliability" and I'm not sure how much it adds to the paper. The age of the basal ice appears to have substantial variability, and I'm not sure that its variability has much to say about the reliability of the model. If you want to explore (as you say) the "assumption [...] that the age–depth function should be rather similar for the same (normalized) ice thickness within a region for small flow velocities and where the overall ice dynamic behaviour (e.g. prevailing divide or flank flow regime) is comparable.", the best way to test this assumption is probably to look at a normalized depth for which your data have good resolution. A normalized depth of 1, where the age is calculated based on a substantial extrapolation from shallower layers, doesn't seem like the right place to do this.*

Thanks for the comment, we have removed this part.

*363: Please provide some detail (an equation, perhaps?) for how delta p was calculated.*

We added an equation for calculating $\Delta X$ ($X$ represents model derived parameter, $X$ could be shape factor, accumulation rate, mechanical ice thickness and age of basal ice).

$$\Delta X = \frac{|X_1 - X_2|}{0.5(X_1 + X_2)}$$

*Table 2 caption: define each variable.*

We defined the variables by changing the caption from " Mean value and standard deviation of relative percentage difference between model runs for the profile 20170240." to " Mean value and standard deviation of relative percentage difference of age of basal ice $\Delta\chi_b$, shape function $\Delta p$, accumulation rate $\Delta\dot{a}$ and mechanical ice thickness $\Delta H_m$ between model runs for the profile 20170240."

*461: should be "…allowed the aircraft to make long-range surveys from a high-altitude camp, covering a large region…"*

We changed "In contrast, despite these shortcomings the simple and light-weight system enabled a long range of the aircraft from a high-altitude field camp to cover a large region around DF." to "In contrast, despite these shortcomings, the simple and light-weight system allowed the aircraft to conduct long-range surveys from a high-altitude field camp, covering a large region around DF."

*469: Please check the sentence beginning "Furthermore," for grammar and parallel structure.*

We changed "Furthermore, although 3-D full Stokes models can lift restrictions, they still come along with new challenges, including heavy computation time, more complicated boundary conditions and conjunction between 3-D model and age observations." to "Furthermore, although 3-D full Stokes models can lift restrictions, they come along with new challenges including heavy computation time and complicated boundary conditions. The conjunction between 3-D model and age observations is still difficult."